# Not All Experts and Tokens Matter: Selective Token-guided Expert Pruning for MoE

## Abstract

Mixture-of-Experts (MoE) architectures achieve exceptional scalability for large language models but present significant deployment challenges due to substantial expert parameter overhead. Existing expert pruning approaches rely on token-agnostic heuristics, such as routing frequency or similar statistical metrics. These methods dilute critical signals from important tokens, conflate statistical presence with functional importance, and completely discard pruned experts' knowledge. To address these limitations, we introduce STEP (Selective Token-guided Expert Pruning), a novel compression framework driven by three key innovations: (i) **Token-aware expert evaluation** that prioritizes important tokens for context-sensitive expert assessment; (ii) **Loss-impact expert scoring** that quantifies expert importance through direct loss contribution rather than statistical proxy metrics; (iii) **Expert-to-bias conversion** that preserves domain knowledge via compact adaptive vectors, transforming pruning from a "discard-and-forget" to a "compress-and-preserve" paradigm. Extensive experiments demonstrate STEP's superiority across model scales and MoE architectures. At 50% expert sparsity of the 30B Qwen model, our pruning method achieves nearly a 50% reduction in memory usage with minimal performance degradation. The method is accompanied by a $1.5\times$ throughput acceleration, and the entire process of pruning and converting the model completes within just 10 minutes. This enables efficient and scalable deployment of MoE models.

## 1 Introduction

The exponential scaling of large language models (LLMs) (Brown et al., 2020; OpenAI et al., 2024) has fundamentally transformed natural language processing, with Mixture-of-Experts (MoE) architectures (Jacobs et al., 1991; Roller et al., 2021) emerging as the dominant paradigm for achieving exceptional performance while maintaining computational efficiency. Through sparse expert activation, contemporary MoE implementations optimize the performance-efficiency trade-off remarkably well. For instance, Qwen3-30B-A3B (abbreviated as Qwen3-30A3B) achieves performance parity with the dense Qwen3-14B model while activating only 3B parameters per token from its 30B parameter set (Yang et al., 2025). However, despite achieving computational efficiency during inference, MoE architectures face fundamental deployment bottlenecks due to massive memory footprints (Yi et al., 2025; Liu et al., 2025a). The Qwen3-30A3B model requires over twice the memory capacity of the 14B dense model for parameter storage, creating substantial barriers in resource-constrained environments. This challenge becomes particularly acute for state-of-the-art models: deploying a 671B parameter model (DeepSeek-AI et al., 2025) demands over 1.3TB of GPU memory, rendering it inaccessible to most practitioners.

Given the infeasibility of full retraining due to proprietary training data and extraordinary computational costs, post-training compression has become essential. Current approaches fall into two categories: **expert matrix compression** methods that compress individual expert parameters (Li et al., 2025b; Chen et al., 2025b), and **expert-level pruning** methods that reduce expert count through removal (Lu et al., 2024; Zhang et al., 2025) or merging (Li et al., 2024; Chen et al., 2025a).

While expert-level pruning provides deployment-friendly compression that aligns well with the trend toward increasingly fine-grained expert architectures, these methods (Koishekenov et al., 2023; He et al., 2024a) suffer from critical limitations. First, they fail to account for token importance, as

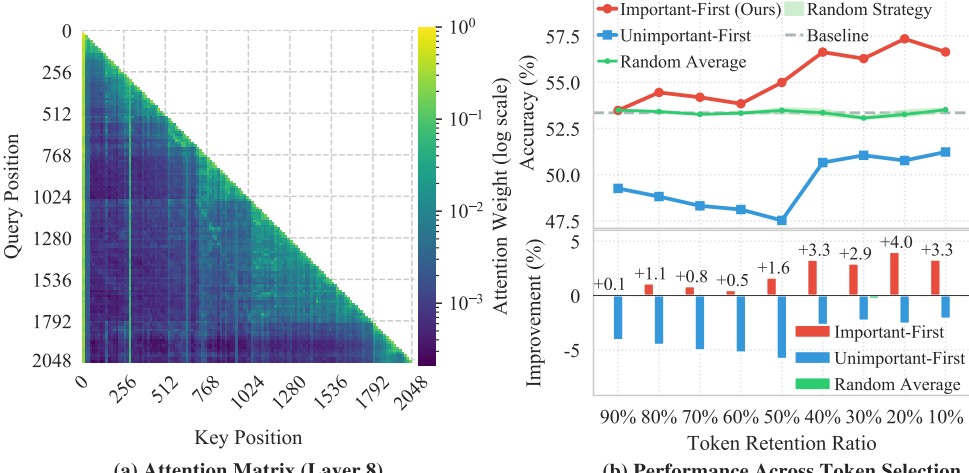

(a) Attention Matrix (Layer 8)    (b) Performance Across Token Selection

Figure 1: Token selection significantly impacts MoE expert pruning effectiveness. (a) Token-to-token attention heatmap from the middle layer of OLMoE-7A1B reveals sparse distribution with concentrated diagonal weights. (b) Performance for frequency-based expert pruning with different token selection strategies. Important-token guidance (red) with higher attention score, achieves up to highest accuracy improvement over the baseline (gray) guided by full-tokens, while unimportant-token guidance (blue), conversely, causes severe performance degradation. Random token selection (green) shows minimal variation around baseline performance, confirming that token selection quality, rather than quantity, is the key determinant of pruning effectiveness.

empirically demonstrated by our analysis of attention map from the middle layer of OLMoE-1B-7B-0125 (abbreviated as OLMoE-7A1B) (Muennighoff et al., 2025), shown in Figure 1. The analysis of the last token's attention distribution reveals highly sparse patterns where only the top 20% of tokens capture approximately 80% of the total attention weights as shown in Appendix H. Correspondingly, across 8 standard datasets (details in Appendix E), frequency-based expert pruning guided by the top 20% important tokens yields up to 4.0% accuracy gains compared to full-token baselines. Second, they rely on statistical proxy metrics across all tokens, which theoretically conflates mere presence with functional significance. Some approaches, such as NAEE (Lu et al., 2024) and EEP (Liu et al., 2024), attempt to account for functional significance through combinatorial search. However, these methods are ill-suited for modern fine-grained MoE due to the combinatorial explosion inherent in their search spaces; and third, they neglect knowledge retention by permanently discarding pruned experts' expertise without preservation mechanisms. These inefficiencies collectively motivate our central research question: **How can we effectively compress MoE models by focusing on expert contributions to truly important tokens while preserving essential knowledge?**

To address these challenges, we propose STEP (Selective Token-guided Expert Pruning), a novel compression framework that transforms MoE optimization from a "discard-and-forget" to a "compress-and-preserve" paradigm. Our method introduces three key innovations:

- **Token-Aware Expert Evaluation**: We develop a plug-and-play method for identifying information-rich tokens based on attention patterns, demonstrating that focusing expert importance assessment on these tokens significantly improves compression quality.

- **Loss-Impact Expert Scoring**: We propose a direct loss-impact metric that quantifies layer-wise loss increases from expert removal, enabling frequency-aware and feature-sensitive importance assessment that moves beyond proxy metrics to performance measurement.

- **Expert-to-Bias Conversion**: Rather than discarding redundant experts entirely, we convert them into compact bias vectors, preserving their knowledge contribution while dramatically reducing parameter overhead, transforming compression from destructive to preservative.

Extensive experiments across model scales (7B to 30B parameters) and three MoE architectures demonstrate STEP's superiority over existing baselines on both language modeling and zero-shot evaluation tasks. On the 30B-parameter Qwen model, STEP completes entire pipeline within merely

10 minutes while maintaining original performance. At 50% pruning ratios, our approach enables immediate hardware acceleration without requiring specialized operators, achieving inference speedups of $1.5\times$ and a nearly 50% reduction in memory usage. These results establish a new compression paradigm for MoEs that enables scalable deployment while preserving model performance.

## 2 RELATED WORK

**Mixture-of-Experts Architectures.** The scaling of large language models has led to prohibitive computational costs, making MoE architectures a practical solution by activating only a subset of experts per token. Starting from early work by Jacobs et al. (Jacobs et al., 1991) and later integration into Transformers by Shazeer et al. (Shazeer et al., 2017), modern MoE systems have achieved unprecedented scale. Recent models like DeepSeek-R1 (DeepSeek-AI, 2025) with 671B parameters, Llama 4 Maverick (Meta, 2025) activating only 17B of 400B parameters, and Qwen3-Next (Qwen Team, 2025) using 3.7% of its parameters demonstrate impressive per-token efficiency but leave most parameters inactive. However, many experts are rarely selected or functionally redundant (Lu et al., 2024; Zhang et al., 2025), creating a critical need for efficient MoE compression techniques.

**MoE-Specific Pruning Methods.** Existing approaches fall into two categories. Matrix-level compression operates on weight matrices within experts, including adapted pruning methods like MoE-Pruner (Xie et al., 2024) and STUN (Lee et al., 2025), shared projection techniques such as Mo-LAE (Liu et al., 2025c) and MoBE (Chen et al., 2025b), and hybrid approaches including $D^2$-MoE (Gu et al., 2025) and ResMoE (Ai et al., 2025). Expert-level pruning directly reduces expert numbers based on routing statistics (Koishekenov et al., 2023; He et al., 2024a), combinations strategy (Lu et al., 2024), dynamic expert replacement as in MoNE (Zhang et al., 2025), and expert merging via clustering demonstrated by MC-SMoE (Li et al., 2024) and HC-SMoE (Chen et al., 2025a). Furthermore, Sub-MoE (Li et al., 2025a) advancing this through adaptive grouping and joint SVD. While these expert-level compression methods effectively address expert proliferation, they critically overlook token importance, a vital dimension for compression efficacy.

**Token Importance-Guided Compression.** Recent research reveals non-uniform token importance in transformers, with certain tokens carrying disproportionately more information while others contribute minimal predictive value (Xiao et al., 2025; Xu et al., 2025). This insight has driven diverse optimization strategies: token pruning in multimodal models via FastV (Chen et al., 2024) and Fit-Prune (Ye et al., 2025), KV cache compression through StreamingLLM (Xiao et al., 2024) and $H_2O$ (Zhang et al., 2023), and quantization integration with RSQ (Sung et al., 2025). Although AdaMoE (Zeng et al., 2024) and MoE++ (Jin et al., 2025) explore token-adaptive routing in MoE architectures, and MC-MoE (Huang et al., 2025) leverages token-guided static bit allocation and online experts skipping like NAEE (Lu et al., 2024), no framework systematically integrates token-importance metrics into expert-pruning methodologies—a critical gap our work addresses.

## 3 METHODOLOGY

This section introduces our method for optimizing sparse Mixture-of-Experts (MoE) models. We first formalize the foundation of MoE architectures, then present an integrated pruning framework combining three complementary techniques to reduce memory footprint while preserving model performance. The complete algorithm workflow is detailed in Algorithm 1.

**Preliminary.** MoE represent a powerful paradigm for scaling neural networks by conditionally activating subsets of parameters. In a standard MoE layer, the computation is governed by:

$$\text{MoE}(x) = \sum_{i \in topk(G(x))} G_i(x) \cdot E_i(x) \tag{1}$$

where $G(x) = \text{softmax}(xW_g)$ represents the gating weights determining expert activation, $E_i(x)$ denotes the $i$-th expert's computation. The gating mechanism routes each input token $x \in \mathbb{R}^d$ to the top-$k$ experts based on learned routing probabilities, where $k$ is the default config of MoE models.

While sparse activation improves computational efficiency, it suffers from significant memory overhead due to the linear growth of expert count $n$: each expert introduces parameter matrices that must be stored regardless of utilization.

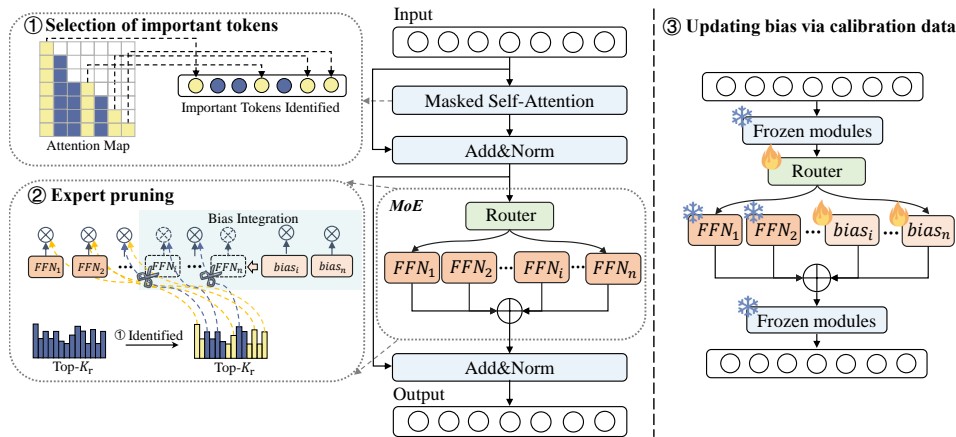

Figure 2: The STEP framework efficiently prunes MoE models via synergistic stages: (1) attention-guided token selection identifies important tokens for pruning; (2) dual-factor expert scoring prunes less critical experts, replacing them with trainable biases; and (3) knowledge-preserving bias updating via calibration data globally, freezing non-essential modules to maintain expressiveness.

**STEP Overview.** Our approach slims existing MoE LLMs through a three-stage framework based on the principle of structured pruning, which guided by selective token and preserves critical expert knowledge while aggressively reducing parameters, as illustrated in Fig 2. The three components work synergistically: attention-guided token selection identifies tokens for pruning decisions, dual-factor expert scoring selects the most critical experts, and knowledge-preserving expert-to-bias conversion preserves information from pruned experts. This integrated design reduces MoE inefficiencies while maintaining model expressiveness.

## 3.1 ATTENTION-GUIDED TOKEN SELECTION

In standard MoE gating, all tokens participate in expert pruning via the router network. This causes a **signal dilution problem**: important tokens that should guide pruning are overwhelmed by numerous less relevant ones, much like informed voters being outnumbered by uninformed ones in an election. Crucially, not all tokens contribute equally to pruning decisions. Many tokens, such as function words or other semantically light tokens, introduce noise rather than signal when determining which computational pathway (expert) should be activated. This noise accumulation leads to suboptimal routing decisions and unnecessary computational overhead.

**Theoretical Motivation.** Expert evaluation often aggregates gating scores across all tokens: $\sum_{t=1}^{T} G_i(x_t)$. We decompose this sum into signal and noise components:

$$\sum_{t=1}^{T} G_i(x_t) = \underbrace{\sum_{t \in \mathcal{T}_{\text{important}}} G_i(x_t)}_{S} + \underbrace{\sum_{t \in \mathcal{T}_{\text{unimportant}}} G_i(x_t)}_{N} \tag{2}$$

The signal-to-noise ratio is $\text{SNR} = \frac{|S|^2}{|N|^2}$. Since unimportant tokens typically outnumber important ones, $|N|$ tends to be large, leading to low SNR and unreliable expert evaluation.

Under effective token selection that keep important tokens while discarding most unimportant ones:

$$\sum_{t \in \mathcal{T}_{\text{selected}}} G_i(x_t) = S + \epsilon \tag{3}$$

where $\epsilon$ represents minimal residual noise from any remaining unimportant tokens. The selected SNR becomes:

$$\text{SNR}_{\text{selected}} = \frac{|S|^2}{|\epsilon|^2} \tag{4}$$

Since $|\epsilon| < |N|$ due to noise reduction, we obtain $\text{SNR}_{\text{selected}} > \text{SNR}_{\text{original}}$. This improvement in signal-to-noise ratio leads to more reliable expert evaluation.

**Implementation Details:** We leverage attention weights to identify important tokens as established by prior work (He et al., 2024b; Chen et al., 2024) that attention mechanisms naturally capture token relevance for downstream computations. For layer $\ell$ with $T$ input tokens, we score token importance $\alpha_t$ using the last token's attention distribution $A_T$:

$$\alpha_t^{(\ell)} = A_{T,t}^{(\ell)} \tag{5}$$

The intuition is that tokens receiving high attention from the last token are more likely to be critical for the final prediction, making them reliable indicators for expert evaluation. The selected token set at layer $\ell$ is formed by selecting the top-$\lceil \tau \cdot T \rceil$ tokens with highest token importance scores:

$$\mathcal{T}_{\text{selected}}^{(\ell)} = \left\{ x_t \mid t \in \arg\text{top-}\lceil \tau \cdot T \rceil \left( A_{T,:}^{(\ell)} \right) \right\} \tag{6}$$

where $\tau$ is the selection ratio and $\mathcal{T}_{\text{selected}}$ is the selected tokens. Expert importance scoring then operates only on selected tokens, this approach concentrates voting power on tokens that reliably indicate which expert is needed, while eliminating noise from unimportant tokens. The selection effectively rebalances voting weights from uniform $\frac{1}{T}$ to concentrated $\frac{1}{|\mathcal{T}_{\text{selected}}|}$, where $|\mathcal{T}_{\text{selected}}| < T$.

## 3.2 Expert importance scoring

Traditional pruning relies on statistics metrics (Koishekenov et al., 2023; Zhuang et al., 2024) that fail to capture the loss impact of experts in MoE architectures. Our dual-factor scoring addresses this by incorporating both expert activation frequency and feature contribution, providing a more nuanced understanding of each expert's role in the model's computational graph. An expert that is rarely activated but produces high-magnitude outputs may be crucial for handling rare yet important cases, whereas a frequently activated expert with consistently low impact may be redundant.

**Mathematical Formulation:** Given an input token $x$, the MoE layer computation explicitly reveals these two critical dimensions:

$$\text{MoE}^{(\ell)}(x) = \underbrace{\sum_{i \in topk\left(G^{(\ell)}(x)\right)}}_{\textbf{Frequency}} \underbrace{G_i^{(\ell)}(x) \cdot E_i^{(\ell)}(x)}_{\textbf{Feature}} \tag{7}$$

To evaluate experts within $\mathcal{T}_{\text{selected}}$, we propose a dual-factor metric combining activation frequency and feature significance which impact MoE layer loss:

$$F_i^{(\ell)} = \frac{1}{|\mathcal{T}_{\text{selected}}^{(\ell)}|} \sum_t \mathbf{1}\left[ G_i^{(\ell)}(x_t) \in \text{top-}k\left(G^{(\ell)}(x_t)\right) \right] \quad \textit{(Activation Frequency)} \tag{8}$$

$$L_i^{(\ell)} = \frac{1}{|\mathcal{T}_{\text{selected}}^{(\ell)}|} \sum_t \left\| G_i^{(\ell)}(x_t) E_i^{(\ell)}(x_t) \right\|_2 \quad \textit{(Feature Norm)} \tag{9}$$

$$\text{Score}(E_i^{(\ell)}) = F_i^{(\ell)} \cdot L_i^{(\ell)} \tag{10}$$

The activation frequency $F_i^{(\ell)}$ measures how often expert $i$ appears among the top-$k$ experts, reflecting its overall utility. The feature norm $L_i^{(\ell)}$ quantifies the magnitude of its contribution to the layer output, weighted by gating probability. Combining these metrics ensures retention of experts that are both consistently useful and impactful. Within each layer, experts are ranked by this composite score, and the lowest-scoring ones are removed according to pruning ratio $p$.

## 3.3 Expert-to-Bias Knowledge Preservation

Since the full computational complexity of an expert is often unnecessary (Lu et al., 2024; Zeng et al., 2024; Jin et al., 2025) and its core contribution for output can be preserved in a lightweight vector (Zhang et al., 2025), we mitigate irreversible information loss from pruning by transforming

essential computational patterns of removed experts into compact bias representations. This plug-and-play bias ($< 0.1\%$ of original weights) retains the expert's knowledge without costly updates.

**Bias Vector Initialization and Integration:** To retain knowledge from expert $i$ in the pruned experts set $\mathcal{R}^{(\ell)}$, we compress its behaviors into bias vectors through activation-weighted averaging:

$$\mathbf{b}_i^{(\ell)} = \frac{1}{|\mathcal{T}_i^{(\ell)}|} \sum_{x_t \in \mathcal{T}_i^{(\ell)}} E_i^{(\ell)}(x_t), \quad \mathcal{T}_i^{(\ell)} = \{x_t \mid G_i^{(\ell)}(x_t) \in \text{top-}k((G^{(\ell)}(x_t)))\} \qquad (11)$$

where $\mathcal{T}_i^{(\ell)}$ denotes the set of selected tokens that highly activate for expert $E_i^{(\ell)}$, $\mathcal{T}_i^{(\ell)} \in \mathcal{T}_{selected}^{(\ell)}$. This averaging process captures the typical output pattern of the expert when it is most relevant, creating a representative bias vector that embodies the expert's specialized computational tendency.

The modified layer output seamlessly integrates these bias vectors. By maintaining the gating structure of the original MoE layer while replacing expensive expert computations with simple bias additions, the formulation becomes:

$$\text{Output}^{(\ell)}(x) = \sum_{j \in topk(G(x))} G_j^{(\ell)}(x) \cdot \left( \mathbf{1}[E_j^{(\ell)} \notin \mathcal{R}^{(\ell)}] E_j^{(\ell)}(x) + \mathbf{1}[E_j^{(\ell)} \in \mathcal{R}^{(\ell)}] \mathbf{b}_j^{(\ell)} \right) \qquad (12)$$

**Bias Vectors Updating:** Bias vectors require a updating phase to integrate smoothly with the remaining active experts in the pruned MoE model $f'$. With this update process being ultra lightweight and requiring only minimal calibration data and compute, the remaining experts in $f'$ assimilate the integrated bias vectors, preserving overall model performance:

$$\min_{\Theta} \sum_{(x,y) \in \mathcal{D}_{\text{cal}}} \mathcal{L}(f'_{\Theta}(x), y) \qquad (13)$$

---

**Algorithm 1** STEP: Selective Token-guided Expert Pruning for MoE LLMs

---

**Require:** MoE model $f$, calibration dataset $\mathcal{D}_{\text{cal}}$, pruning ratio $p$, token retain threshold $\tau$
**Ensure:** Pruned $f'$ with preserved performance
1: **for** layer $\ell = 1$ **to** $L$ **do**
2: $\quad \mathcal{T}_{selected}^{(\ell)} \leftarrow \left\{ x_t \mid t \in \arg \text{topk}_{\lceil \tau \cdot T \rceil} \left( A_{T,:}^{(\ell)} \right) \right\}$ {**Stage 1. Token Selection**}
3: $\quad$ **for** each expert $E_i^{(\ell)}$ **do**
4: $\quad\quad F_i^{(\ell)} \leftarrow \frac{1}{|\mathcal{T}_{selected}^{(\ell)}|} \sum_t \mathbf{1}[G_i^{(\ell)}(x_t) \in \text{top-}k \left( G^{(\ell)}(x_t) \right)]$
5: $\quad\quad L_i^{(\ell)} \leftarrow \frac{1}{|\mathcal{T}_{selected}^{(\ell)}|} \sum_t \|G_i^{(\ell)}(x_t) \cdot E_i^{(\ell)}(x_t)\|_2^2$
6: $\quad\quad \text{Score}_i^{(\ell)} \leftarrow F_i^{(\ell)} \cdot L_i^{(\ell)}$
7: $\quad$ **end for**
8: $\quad \mathcal{R}^{(\ell)} \leftarrow \{E_i^{(\ell)} \mid \text{Score}_i^{(\ell)} \in \text{lowest } p\%\}$ {**Stage 2. Expert Importance Scoring**}
9: $\quad$ **for** $E_i^{(\ell)} \in \mathcal{R}^{(\ell)}$ **do**
10: $\quad\quad \mathbf{b}_i^{(\ell)} \leftarrow \frac{1}{|\mathcal{T}_i^{(\ell)}|} \sum_{x_t} E_i^{(\ell)}(x_t)$ where $\mathcal{T}_i^{(\ell)} = \{x_t \mid G_i^{(\ell)}(x_t) \in \text{top-}k((G^{(\ell)}(x_t)))\}$
11: $\quad$ **end for**
12: $\quad$ Remove $\mathcal{R}^{(\ell)}$ from $f^{(\ell)}$
13: **end for**
14: Modify $f'$: $\text{Output}^{(\ell)} \leftarrow \sum_j G_j^{(\ell)}(x) \cdot \left( \mathbf{1}[E_j^{(\ell)} \notin \mathcal{R}^{(\ell)}] E_j^{(\ell)}(x) + \mathbf{1}[E_j^{(\ell)} \in \mathcal{R}^{(\ell)}] \mathbf{b}_j^{(\ell)} \right)$
15: $\Theta^* \leftarrow \arg \min_{\Theta} \sum_{(x,y) \in \mathcal{D}_{\text{cal}}} \mathcal{L}(f'_{\Theta}(x), y)$ {**Stage 3. Expert to Bias Converting**}
16: **return** $f'_{\Theta^*}$

---

## 4 EXPERIMENTS

We evaluate STEP on four dimensions: zero-shot task, language modeling, ablation analysis, and computational efficiency. Experiments setup and baselines are detailed in Appendix E.

### 4.1 ZERO-SHOT TASK PERFORMANCE

Table 1: Zero-Shot Task Accuracy(%) Comparison Across Models Under Structured Pruning at 25% and 50% MoE Pruning Ratios. m experts with n% density denoted as $m_{n\%}$, Bold indicates best performance, underlined indicates second-best.

| Model | Experts | Method | ARC-E | ARC-C | BoolQ | PIQA | Wino | Hella | MMLU | OBQA | Avg |
|---|---|---|---|---|---|---|---|---|---|---|---|
| | 64 | Original | 77.23 | 46.84 | 70.15 | 78.62 | 69.06 | 58.44 | 53.54 | 44.20 | 62.26 |
| | $64_{75\%}$ | D²-MoE | 71.21 | 37.46 | 66.67 | 75.35 | 66.85 | 49.63 | 28.39 | 41.20 | 54.60 |
| | | MC-SMoE | 70.12 | 38.91 | 52.26 | 66.32 | 61.96 | 41.47 | 37.83 | 35.90 | 50.60 |
| | | HC-SMoE | 71.25 | 39.59 | 65.78 | 72.42 | 66.22 | 51.69 | 42.65 | 29.40 | 54.88 |
| | 48 | GS | 54.88 | 28.58 | 62.11 | 73.12 | 63.30 | 50.22 | 24.50 | 39.90 | 49.58 |
| | | MoNE | 62.37 | 33.28 | 65.84 | 77.09 | 67.32 | 54.88 | 23.27 | 42.00 | 53.26 |
| OLMoE-7A1B | | **Ours** | **73.82** | **44.03** | **68.35** | **77.53** | 65.51 | **55.24** | 39.70 | **43.20** | **58.42** |
| | $64_{50\%}$ | D²-MoE | 60.37 | 29.36 | 62.03 | 69.64 | 58.93 | 39.70 | 25.81 | 29.20 | 46.88 |
| | | MC-SMoE | 30.81 | 18.52 | 42.66 | 55.60 | 50.36 | 27.41 | 23.00 | 26.90 | 34.41 |
| | | HC-SMoE | 46.84 | 26.28 | 58.20 | 62.19 | 59.12 | 39.08 | 29.60 | 28.20 | 43.69 |
| | 32 | GS | 34.93 | 19.03 | 43.85 | 59.47 | 50.83 | 31.03 | 23.55 | 23.70 | 35.80 |
| | | MoNE | 48.91 | 24.74 | 62.39 | 71.11 | 59.67 | 44.96 | 23.10 | 31.60 | 45.81 |
| | | **Ours** | **64.10** | **31.40** | **63.52** | **73.88** | 58.72 | **46.17** | 26.40 | **36.20** | **50.05** |
| | 64 | Original | 84.51 | 56.14 | 80.37 | 78.94 | 71.11 | 59.27 | 67.29 | 45.00 | 67.83 |
| | $64_{75\%}$ | D²-MoE | 78.20 | 45.22 | 74.80 | 76.55 | 69.38 | 49.89 | **53.74** | 42.20 | 61.25 |
| | | MC-SMoE | 77.36 | 45.22 | 79.08 | 78.40 | **72.14** | 56.15 | 49.00 | 33.60 | 61.37 |
| | | HC-SMoE | 69.32 | 37.20 | 69.11 | 68.99 | 59.12 | 40.00 | 52.55 | 34.80 | 53.89 |
| | 48 | GS | 78.45 | 48.12 | 77.58 | 79.71 | 70.96 | 58.52 | 50.66 | 45.20 | 63.65 |
| | | MoNE | 80.89 | 50.94 | 78.53 | 80.03 | 70.72 | **58.88** | 49.25 | 45.40 | 64.33 |
| Moonlight-16A3B | | **Ours** | **82.17** | **52.29** | **79.17** | **80.10** | 71.03 | 58.88 | 52.62 | **45.90** | **65.27** |
| | $64_{50\%}$ | D²-MoE | 65.57 | 33.96 | 67.74 | 69.80 | 61.96 | 41.25 | **32.92** | 33.80 | 50.88 |
| | | MC-SMoE | 62.58 | 28.24 | 63.24 | 68.33 | 59.51 | 38.97 | 22.90 | 27.20 | 46.37 |
| | | HC-SMoE | 53.54 | 24.74 | 60.89 | 62.30 | 50.99 | 30.68 | 30.81 | 30.00 | 42.99 |
| | 32 | GS | 62.37 | 31.57 | 54.68 | 72.80 | 63.30 | 46.17 | 22.92 | 36.20 | 48.75 |
| | | MoNE | 66.75 | 33.79 | 71.13 | 77.48 | 70.56 | 53.06 | 22.97 | **40.00** | 54.47 |
| | | **Ours** | **71.14** | **38.48** | **74.09** | **77.64** | 68.14 | **54.40** | 25.87 | 39.80 | **56.20** |
| | 128 | Original | 79.84 | 53.24 | 88.62 | 79.38 | 70.64 | 59.51 | 77.79 | 44.60 | 69.20 |
| | $128_{75\%}$ | D²-MoE | **80.56** | 54.01 | 88.56 | 79.43 | 70.17 | 59.61 | **73.78** | 44.00 | 68.77 |
| | | MC-SMoE | 64.69 | 39.16 | 77.65 | 67.08 | 58.80 | 41.45 | 54.00 | 31.80 | 54.33 |
| | | HC-SMoE | 75.97 | 45.82 | 86.54 | 76.22 | 70.40 | 51.54 | 66.89 | 41.00 | 64.30 |
| | 96 | GS | 77.99 | 50.77 | 88.50 | 79.43 | 69.93 | 59.05 | 72.77 | 42.60 | 67.63 |
| | | MoNE | 80.35 | 54.27 | **89.11** | 79.87 | 70.48 | 59.52 | 72.83 | 44.20 | 68.83 |
| Qwen3-30A3B | | **Ours** | 80.43 | **54.69** | 88.83 | **79.98** | **70.80** | **59.84** | 72.90 | **45.00** | **69.06** |
| | $128_{50\%}$ | D²-MoE | 66.12 | 40.19 | 79.51 | 75.03 | 64.09 | 53.37 | 34.06 | 38.00 | 56.30 |
| | | MC-SMoE | 30.89 | 19.45 | 58.32 | 55.11 | 46.96 | 27.32 | 26.08 | 25.00 | 36.14 |
| | | HC-SMoE | 65.19 | 35.84 | 83.03 | 70.95 | 65.19 | 41.84 | 46.06 | 36.20 | 55.54 |
| | 64 | GS | 48.40 | 28.07 | 82.23 | 69.64 | 61.88 | 48.29 | 42.00 | 32.80 | 51.66 |
| | | MoNE | 76.56 | 46.76 | 87.61 | 79.54 | 69.85 | 57.97 | 51.67 | 43.20 | 64.15 |
| | | **Ours** | **78.58** | **47.70** | **88.04** | **79.61** | **70.33** | **58.24** | **58.72** | **45.00** | **65.78** |

Our method demonstrates consistent superiority across zero-shot task generalization with notable gains in average accuracy over existing approaches as detailed in Table 1. On OLMoE-7A1B at 25% sparsity, we achieve an average accuracy of 58.42, outperforming HC-SMoE by 3.54% and MoNE by 5.16%. This improvement is reflected across diverse benchmarks where our approach either leads or remains competitive, consistently ranking first or second on individual datasets. Furthermore, scalability is evident in experiments conducted on Qwen3-30A3B with 64 experts, where our method attains an average accuracy of 65.78%, exceeding MoNE by 1.63% even under 50% expert reduction. Performance evaluation across varying levels of expert granularity further substantiates the robustness of our approach. As illustrated in Figure 3, when the number of experts is reduced from 64 to 20, our method consistently outperforms baselines

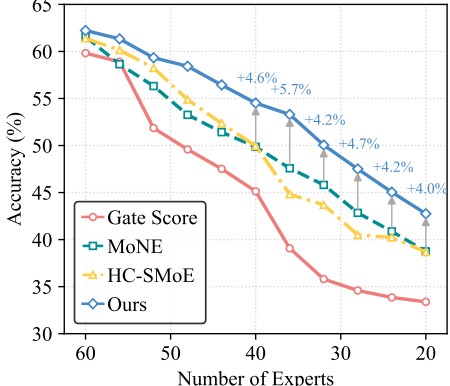

Figure 3: Average accuracy of OLMoE-7A1B across 8 benchmarks with varying expert numbers and pruning methods.

with progressively widening margins at higher compression ratios. Notably, during the transition from 64 to 20 experts, the maximum improvement in average accuracy reaches 5.7%, demonstrating the sustained advantage of our method across pruning ratios.

### 4.2 LANGUAGE MODELING PERFORMANCE

Table 2: Language Modeling Performance (Perplexity↓) Comparison Across Models Under Structured Pruning: OLMoE-7A1B, Moonlight-16A3B, and Qwen3-30A3B at 25% and 50% MoE Pruning Ratios. m experts with n% density denoted as $m_{n\%}$, Gray cells highlight the best results.

| Model | Method | Experts | C4 | Wiki | Avg | Experts | C4 | Wiki | Avg |
|---|---|---|---|---|---|---|---|---|---|
| | Original | 64 | 12.14 | 6.65 | 9.40 | 64 | 12.14 | 6.65 | 9.40 |
| OLMoE-7A1B | $D^2$-MoE | $64_{75\%}$ | 8.83e3 | 1.66e4 | 1.27e4 | $64_{50\%}$ | 1.19e4 | 2.26e4 | 1.73e4 |
| | MC-SMoE | 48 | 20.50 | 11.09 | 15.80 | 32 | 539.21 | 407.46 | 473.34 |
| | HC-SMoE | 48 | 17.07 | 12.15 | 14.61 | 32 | 39.39 | 31.45 | 35.42 |
| | GS | 48 | 17.12 | 16.90 | 17.01 | 32 | 109.08 | 183.63 | 146.36 |
| | MoNE | 48 | 15.63 | 12.23 | 13.93 | 32 | 32.00 | 48.11 | 40.06 |
| | **Ours** | 48 | **14.76** | **10.40** | **12.58** | 32 | **23.33** | **24.60** | **23.97** |
| | Original | 64 | 11.30 | 7.12 | 9.21 | 64 | 11.30 | 7.12 | 9.21 |
| Moonlight-16A3B | $D^2$-MoE | $64_{75\%}$ | 16.09 | 11.01 | 13.55 | $64_{50\%}$ | 30.14 | 23.62 | 26.88 |
| | MC-SMoE | 48 | 12.96 | 7.80 | 10.38 | 32 | 43.52 | 25.70 | 34.61 |
| | HC-SMoE | 48 | 33.32 | 17.99 | 25.66 | 32 | 156.30 | 75.73 | 116.02 |
| | GS | 48 | 11.86 | 7.33 | 9.60 | 32 | 27.86 | 18.61 | 23.24 |
| | MoNE | 48 | 11.72 | 7.27 | 9.50 | 32 | 18.47 | 12.17 | 15.32 |
| | **Ours** | 48 | **11.69** | **7.27** | **9.48** | 32 | **14.93** | **10.07** | **12.50** |
| | Original | 128 | 14.06 | 8.70 | 11.38 | 128 | 14.06 | 8.70 | 11.38 |
| Qwen3-30A3B | $D^2$-MoE | $128_{75\%}$ | 14.51 | 8.84 | 11.68 | $128_{50\%}$ | 32.78 | 25.62 | 29.20 |
| | MC-SMoE | 96 | 25.53 | 16.84 | 21.19 | 64 | 217.86 | 123.05 | 170.46 |
| | HC-SMoE | 96 | 18.25 | 12.52 | 15.39 | 64 | 28.44 | 20.60 | 24.52 |
| | GS | 96 | 14.92 | 8.90 | 11.91 | 64 | 36.27 | 26.34 | 31.31 |
| | MoNE | 96 | 14.39 | 8.93 | 11.66 | 64 | 16.36 | 11.36 | 13.86 |
| | **Ours** | 96 | **13.63** | **8.07** | **10.85** | 64 | **15.06** | **9.81** | **12.44** |

STEP consistently achieves superior perplexity preservation across all model scales and sparsity levels. Our method outperforms existing baselines by a notable margin, delivering up to 27% relative perplexity reduction compared to MoNE on OLMoE-7B at 50% sparsity. In contrast, expert merging techniques such as MC-SMoE and HC-SMoE suffer from significant performance degradation, highlighting the importance of preserving rather than merging expert knowledge. The gap widens under more aggressive pruning ratio. As shown in Appendix F.1, our method's perplexity increases linearly while others collapse exponentially, demonstrating the robustness of our approach.

### 4.3 ABLATION ANALYSIS

Table 3: Ablation study on three core components with 25% experts pruned on OLMoE-7A1B, and the hyper-parameter $\tau$ of token selection is set to 0.5. Note: ✓ indicates inclusion.

| Method Components | | | Zero-shot Tasks Accuracy(%) | | | | | | | | Average |
|---|---|---|---|---|---|---|---|---|---|---|---|
| Token Selection | Feature Norm | Experts-to-Bias | ARC-E | ARC-C | PIQA | Wino | BoolQ | Hella | MMLU | OBQA | |
| | | | 62.37 | 34.04 | 75.89 | 67.32 | 62.94 | 53.11 | 31.13 | 41.00 | 53.48 |
| ✓ | | | 62.04 | 37.29 | 75.25 | 66.55 | 67.62 | 53.65 | 37.33 | 41.20 | 55.12 |
| | ✓ | | 66.28 | 37.37 | 77.26 | 67.56 | **68.56** | **55.89** | 29.11 | 41.40 | 55.43 |
| | ✓ | ✓ | 68.79 | 39.77 | 77.48 | **67.64** | 67.33 | 55.65 | 30.44 | 42.20 | 56.16 |
| ✓ | ✓ | | 71.30 | 42.40 | 77.42 | 65.27 | 68.14 | 55.21 | 38.14 | 42.20 | 57.51 |
| ✓ | ✓ | ✓ | **73.82** | **44.03** | **77.53** | 65.51 | 68.35 | 55.24 | **39.70** | **43.20** | **58.42** |

We conduct ablation studies incrementally augmenting a frequency-based baseline with core components. As detailed in Table 3, incorporating **token selection** yields +1.64% average gain versus baseline. Combining **feature norm** with **expert-to-bias** achieves +2.58% improvement, while adding token importance to this configuration provides an additional +2.26% boost. Full integration delivers optimal performance, demonstrating component synergy. We further study **calibration robustness** to assess method stability across different data scales. All evaluated methods maintain perplexity variance below 0.3 and ARC-C accuracy variance under 1% across sample sizes from 32 to 512 sequences, except for MC-SMoE, which exhibits substantially higher variances of 7.4 and 5.1%, respectively, as shown in Figure 5. This comparative analysis confirms our method's insensitivity to calibration data scale, thereby enhancing practical deployability. Hyperparameter ablations (**pruning ratio**, **token selection ratio**, **training epochs**) in Appendix F confirm robustness.

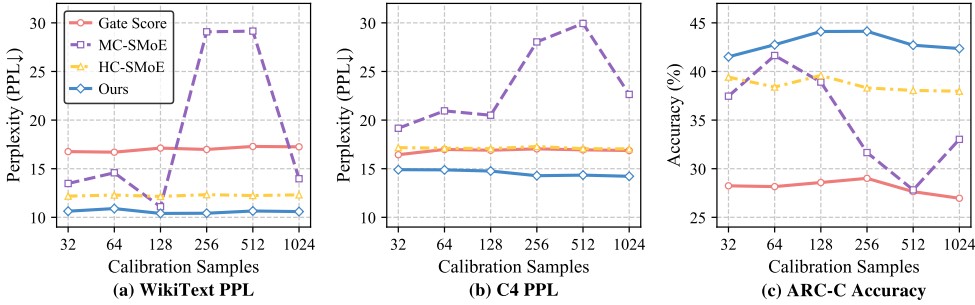

Figure 5: Ablation Study on Calibration Stability: Performance of OLMoE-7A1B under varying numbers of calibration samples and different pruning methods.

## 4.4 RESOURCE CONSUMPTION AND SPEEDUP

To assess the practical efficiency of our pruning algorithm, we evaluate computational overhead, memory usage, and inference performance across multiple architectures and pruning ratios, using a consistent token configuration (1024 for prefill, 256 for decoding). Empirical results from Table 4 and Figure 4 on Qwen3-30A3B show significant improvements: our method reduces memory footprint and parameter count by 23.4%–47.5%, while achieving 1.10–1.50× inference speedup in both prefill and decoding stages. Additionally, the entire process is completed within 10 minutes with a negligible peak memory increase (e.g., 59.84GB vs. 57.09GB), incurring minimal resource overhead. The linear correlation between pruning ratio and resource reduction underscores the predictability and deployment suitability of our approach under constrained resource budgets. Further evaluation on OLMoE-7A1B model, detailed in Appendix G, confirms the generalizability of efficiency gains.

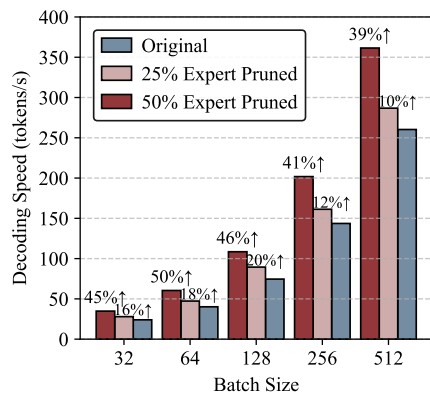

Figure 4: Throughput of Qwen3-30A3B under different expert pruning ratios and different batch sizes.

Table 4: Model Prefill Performance and Resource Consumption Analysis

| Model | Experts | Resource Consumption | | | | Prefill Performance | | Pruning Overhead | | |
| | | Memory (GB) | | Parameters (B) | | Latency | Speedup | Time(mins) | | Memory |
| | | Value | ↓% | Value | ↓% | (ms) | | Prune | Bias | (GB) |
|---|---|---|---|---|---|---|---|---|---|---|
| **Qwen3-30A3B** | 128 | 57.09 | 0.0 | 30.53 | 0.0 | 183.14 | 1.00× | – | – | – |
| | 96 | 43.46 | 23.9 | 23.29 | 23.7 | 164.64 | 1.11× | 3 | 7 | 59.84 |
| | 64 | 29.96 | 47.5 | 16.04 | 47.5 | 139.49 | 1.31× | | | |

## 5 CONCLUSION

As Mixture-of-Experts (MoE) architectures represent an emerging trend in scaling LLMs with superior performance at equivalent computational cost, their substantial memory requirements pose significant challenges to practical deployment. In this paper, we propose STEP, a token-guided MoE compression method that pioneers the application of token importance to MoE pruning, with theoretical analysis validating its effectiveness. Furthermore, our work marks a paradigm shift from "discard-and-forget" to "compress-and-preserve" pruning, enabling the partial retention of information from pruned experts through expert-to-bias conversion. Comprehensive experiments demonstrate that STEP significantly outperforms existing MoE pruning approaches across various metrics and model scales. Notably, applying STEP to the Qwen-30B model achieves nearly 50% memory compression and a 1.5× speedup while incurring only minimal performance degradation, completing the entire pruning process within 10 minutes. These outcomes enable low-cost and economical deployment of MoE models in practical applications.

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

## APPENDIX

This appendix provides comprehensive supplementary materials: a statement on large language model usage protocols (Section A); a statement on research ethics (Section B); a statement on reproducibility (Section C); analysis of research limitations and future directions including scalability challenges and multi-modal adaptation constraints (Subsections D.1 and D.2); detailed experimental configurations and baseline specifications (Section E); extended hyperparameter ablation (Section F); analysis of model inference performance and resource consumption of OLMoE-7A1B (Section G); and comparative visualization of layer-wise attention distributions (Section H).

## A  LLM USAGE STATEMENT

Large language models (LLMs) were employed **solely** for linguistic refinement in this work, including grammar correction, syntax improvement, and terminology consistency enhancement. The LLM played **no role** in research ideation, methodology design, data analysis, or substantive content generation. All intellectual contributions originate from the human authors, who take full responsibility for the manuscript's content and integrity.

## B  ETHICS STATEMENT

Our research was conducted in accordance with the ICLR Code of Ethics and the general principles of responsible AI research. This work seeks to contribute positively to scientific progress and human well-being. We ensured that no harm was caused during the study and that no human or animal subjects were involved. All datasets used in this work were collected and shared in compliance with their respective licenses and guidelines, with careful attention to privacy protection and confidentiality. We strived to minimize potential biases and ensure fairness in both data selection and evaluation. Transparency, honesty, and reproducibility guided our methodology throughout the research process.

## C  REPRODUCIBILITY STATEMENT

To facilitate reproducibility, we carefully document all components of our study. The datasets employed in our experiments are publicly available, and the models we utilize are open-source. Our methodology, including the overall pipeline, training procedures, default hyperparameter configurations, and ablation study settings, is comprehensively described in the main text and appendix. These detailed descriptions are intended to enable independent researchers to replicate our results and verify our findings with minimal ambiguity.

## D  LIMITATIONS AND FUTURE WORK

While STEP demonstrates significant improvements in MoE model compression, several limitations warrant further investigation, which we outline below alongside proposed future directions.

### D.1  LARGE-SCALE MoE PRUNING EXPERIMENTS

Due to computational constraints, our experiments were conducted on mid-scale MoE architectures (Qwen3-30A3B (Yang et al., 2025) and Moonlight 16A3B[1] (Liu et al., 2025b)) rather than state-of-the-art giants like Qwen3-235B-A22B (Yang et al., 2025) or DeepSeekV3-671B MoE (DeepSeek-AI et al., 2025). While our method shows consistent gains across tested scales (§4), the extrapolation of token-aware pruning efficacy to 100B+ parameter regimes remains unverified. Future work will validate STEP's scalability on trillion-parameter MoEs, with emphasis on dynamic resource allocation strategies for ultra-large models.

---

[1]Same architecture as DeepSeek-V3.

## D.2 EXTENSION TO MULTI-MODAL ARCHITECTURES

Our study focuses exclusively on *language-only* MoE models. Vision-language models (VLMs) present unique challenges: the inherent disparity in token importance distributions between visual patches and linguistic tokens (Chen et al., 2024; Ye et al., 2025) may amplify the limitations of routing-frequency heuristics criticized in §1. Given that STEP's token-aware evaluation (§3) explicitly prioritizes informatively critical tokens, we hypothesize its impact could be *more pronounced* in VLMs where modality-specific token significance varies radically. Future research will extend our framework to multi-modal settings, examining cross-modal token importance fusion.

# E EXPERIMENTAL SETUP AND BASELINES

## E.1 EXPERIMENTAL SETUP

For calibration, we follow established pruning practices (Sun et al., 2024; Lu et al., 2024) utilizing 128 randomly sampled sequences (2,048 tokens each) from the C4 dataset (Raffel et al., 2020), providing sufficient linguistic diversity for robust expert importance estimation.

Three representative MoE architectures spanning different scales were evaluated: the compact OLMoE-7A1B[2] (7B parameters, activates 1B parameters, 64 experts) (Muennighoff et al., 2025), medium-scale Moonlight-16A3B[3] (16B parameters, activates 3B parameters, 64 experts) (Liu et al., 2025b), and large-scale Qwen3-30A3B[4] (30B parameters, activates 3B parameters, 128 experts) (Yang et al., 2025). All models were implemented via Hugging Face Transformers (Wolf et al., 2020) using official configurations, and all the experiments could run on a single A100 GPU.

Evaluation encompassed both intrinsic language modeling performance (perplexity on C4 Raffel et al. (2020) and WikiText (Merity et al., 2016)) and extrinsic zero-shot reasoning capabilities across eight diverse benchmarks from the lm_eval framework (Gao et al., 2021): PIQA (Bisk et al., 2020), BoolQ (Clark et al., 2019), HellaSwag (Zellers et al., 2019), WinoGrande (Sakaguchi et al., 2019), ARC-Easy/Challenge (Clark et al., 2018), OpenBookQA (Mihaylov et al., 2018), and MMLU (Hendrycks et al., 2021).

## E.2 BASELINE COMPARISONS

STEP was rigorously compared against state-of-the-art MoE compression approaches across three methodological categories: (1) **Intra-expert compression** represented by $D^2$-MoE (Gu et al., 2025) (delta decompression only, excluding its structured pruning on merged base weights for fair comparison); (2) **Expert pruning** including Gate Score (GS) based pruning (He et al., 2024a) and state-of-the-art MoNE (Zhang et al., 2025) which converts redundant experts into lightweight novices; (3) **Expert merging** featuring MC-SMoE (Li et al., 2024) (routing-guided expert merging with decomposition) and HC-SMoE (Chen et al., 2025a) (hierarchical feature clustering). All baselines were evaluated under identical sparsity configurations.

# F ADDITIONAL ABLATION STUDY ON KEY HYPERPARAMETERS

## F.1 PRUNING PERFORMANCE ANALYSIS ACROSS EXPERT NUMBERS

Figure 6 presents a comprehensive comparison of perplexity (PPL) results across different pruning methods with varying numbers of experts on both C4 and WikiText datasets. The expert count ranges from 64 to 20 (step size of 4), representing different levels of model sparsity. Our pruning method achieves superior performance preservation at high pruning ratios, significantly outperforming baseline approaches. At a 68.75% pruning ratio (reducing experts from 64 to 20), it reduces perplexity by approximately 98% compared to GS, 72% compared to MONE, and 77% compared to HC-SMoE on WikiText, with similar improvements of 99% over GS, 62% over MONE, and 83%

---

[2]https://huggingface.co/allenai/OLMoE-1B-7B-0125

[3]https://huggingface.co/moonshotai/Moonlight-16B-A3B

[4]https://huggingface.co/Qwen/Qwen3-30B-A3B

over HC-SMoE on C4. Performance degradation remains nearly linear and graceful as pruning intensity increases, while other methods exhibit exponential drops. This consistent superiority across datasets demonstrates robust generalization, with relative gains amplifying at higher pruning levels.

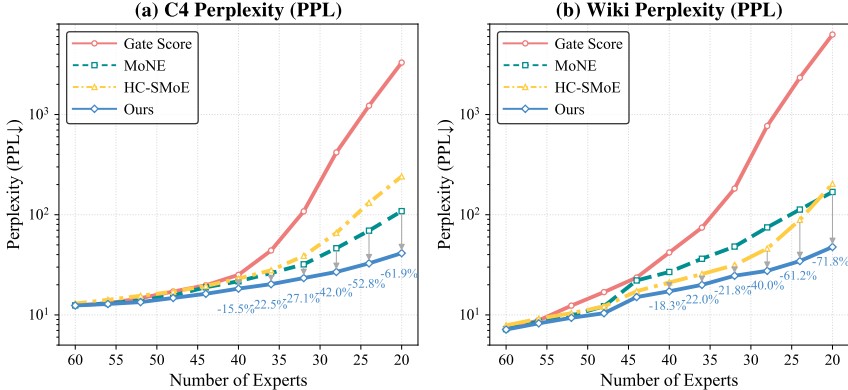

Figure 6: Perplexity of OLMoE-7A1B with varying expert numbers and pruning methods.

## F.2 TOKEN SELECTION RATIO

Table 5: Ablation Study on Token Selection Ratio with 25% Expert Pruning on OLMoE-7A1B

| Ratio | Perplexity | | | Zero-shot Accuracy (%) | | | | | | | | |
|---|---|---|---|---|---|---|---|---|---|---|---|---|
| | Wiki | C4 | Avg | ARC-E | ARC-C | BoolQ | PIQA | Wino | Hella | MMLU | OBQA | Avg |
| 1.0 | 11.65 | 14.52 | 13.09 | 68.79 | 39.77 | 67.34 | 77.48 | 67.64 | 55.65 | 30.44 | 42.20 | 56.16 |
| 0.9 | 11.47 | 14.79 | 13.13 | 69.32 | 39.16 | 64.65 | 77.42 | 67.88 | 55.56 | 32.80 | 41.40 | 56.02 |
| 0.8 | 10.75 | 14.60 | 12.68 | 71.09 | 41.38 | 67.03 | 77.31 | **68.51** | 55.41 | 33.72 | 41.20 | 56.96 |
| 0.7 | 10.80 | 14.89 | 12.85 | 72.39 | 42.84 | 70.40 | 77.86 | 66.38 | 55.97 | 38.19 | 43.20 | 58.40 |
| 0.6 | 10.52 | 14.96 | 12.74 | 72.73 | 41.98 | 71.01 | 77.75 | 66.85 | 55.45 | 38.66 | 41.80 | 58.28 |
| 0.5 | 10.40 | **14.76** | **12.58** | 73.82 | **44.03** | 68.35 | 77.53 | 65.52 | 55.24 | 39.70 | **43.20** | 58.42 |
| 0.4 | 10.55 | 15.04 | 12.80 | 74.07 | 43.52 | 67.92 | 78.02 | 65.04 | 55.40 | 40.60 | 42.60 | 58.39 |
| 0.3 | 10.30 | 15.40 | 12.85 | 74.75 | 43.09 | 67.52 | **78.07** | 64.72 | 55.12 | 42.46 | 41.00 | 58.34 |
| 0.2 | **10.10** | 15.32 | 12.71 | **76.01** | 43.17 | **70.52** | 77.58 | 64.25 | 54.48 | **42.25** | 42.20 | **58.81** |
| 0.1 | 10.30 | 15.56 | 12.93 | 75.67 | 41.30 | 67.31 | 77.15 | 63.06 | 54.07 | 42.31 | 42.80 | 57.96 |

Token selection ratio 0.5 is established as the default configuration, achieving optimal perplexity (12.58) while maintaining competitive zero-shot performance (58.42%), as detailed in Table 5. This ratio balances language modeling fundamentals with zero-shot task capabilities, yielding best-in-class results on ARC-C (44.03%) and OBQA (43.20%). Although ratio 0.2 yields a marginally higher zero-shot average (58.81%), it incurs a 0.8% relative perplexity degradation, favoring 0.5's superior stability-performance equilibrium.

## F.3 BIAS UPDATING HYPERPARAMETER CONFIGURATION

Table 6: Hyperparameter Specifications

| Parameter | Value |
|---|---|
| Optimizer | AdamW |
| Learning rate | $1 \times 10^{-4}$ |
| Weight decay | 0.1 |
| Epsilon ($\epsilon$) | $1 \times 10^{-6}$ |
| Precision | bfloat16 |
| Training steps | 128 |

Table 7: Perplexity by Training Epoch with 25% Expert Pruning on OLMoE-7A1B

| Epochs | C4 | Wiki |
|---|---|---|
| 0 | 15.52 | 12.87 |
| 1 | 14.76 | 10.40 |
| 3 | 14.53 | 10.37 |
| 10 | 14.43 | 10.98 |

Table 6 details the hyperparameter specifications for bias updating. As shown in Table 7, perplexity decreases rapidly during the initial training phase, with the majority of gains achieved within the

first epoch. Subsequent epochs yield diminishing returns, with identical results observed at epochs 3 (14.53/10.37) and 10 (14.43/10.98). This indicates model convergence occurs early in training. We therefore adopt the 1-epoch configuration as the optimal balance between performance and computational efficiency for bias updating.

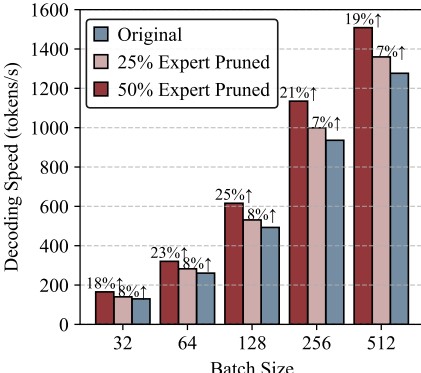

Figure 7: Throughput and Speedup of OLMoE-7A1B under different expert pruning ratios and different batch sizes.

## G  MODEL INFERENCE PERFORMANCE AND RESOURCE CONSUMPTION ANALYSIS OF OLMOE-7A1B

To further validate the efficiency of our pruning method across different architectures, we conduct experiments on the OLMoE-7A1B model. Figure 7 illustrates the throughput and speedup under varying expert pruning ratios and batch sizes, demonstrating consistent performance gains. Table 8 summarizes resource consumption and prefill performance, showing up to 46.6% memory reduction and 1.24× speedup with minimal pruning overhead.

Table 8: Model Prefill Performance and Resource Consumption Analysis of OLMoE-7A1B.

| Model | Experts | Resource Consumption | | | | Prefill Performance | | Pruning Overhead | | |
|---|---|---|---|---|---|---|---|---|---|---|
| | | Memory (GB) | | Parameters (B) | | Latency | Speedup | Time(mins) | | Memory |
| | | Value | ↓% | Value | ↓% | (ms) | | Prune | Bias | (GB) |
| **OLMoE-7A1B** | 64 | 12.91 | 0.0 | 6.92 | 0.0 | 43.07 | 1.00× | – | – | – |
| | 48 | 9.89 | 23.4 | 5.31 | 23.3 | 39.20 | 1.10× | | | |
| | 32 | 6.90 | 46.6 | 3.70 | 46.5 | 34.69 | 1.24× | 1 | 2 | 13.88 |

## H  ATTENTION MAP ACROSS DIFFERENT LAYERS

As demonstrated in Figure 8 and Figure 9, the token-to-token attention heatmap across layers in OLMoE-7A1B exhibits three key characteristics: (i) Sparse weight distribution with most attention weights below $10^{-3}$, (ii) Diagonal concentration of dominant weights, and (iii) Pronounced attention sink at initial token positions.

Quantitatively, for the last token's attention distribution, the top 20% tokens account for $\sim 80\%$ of cumulative attention weight, with layer-wise variations:

- **Layer 16**: Top 20% tokens account for 92.7% attention weight
- **Layer 12**: Top 20% tokens account for 93.8% attention weight
- **Layer 8**: Top 20% tokens account for 93.4% attention weight
- **Layer 4**: 75.5% concentration, the minimum observed value

These findings underscore that optimal expert pruning strategies must account for layer-wise variations in token selection dynamics.

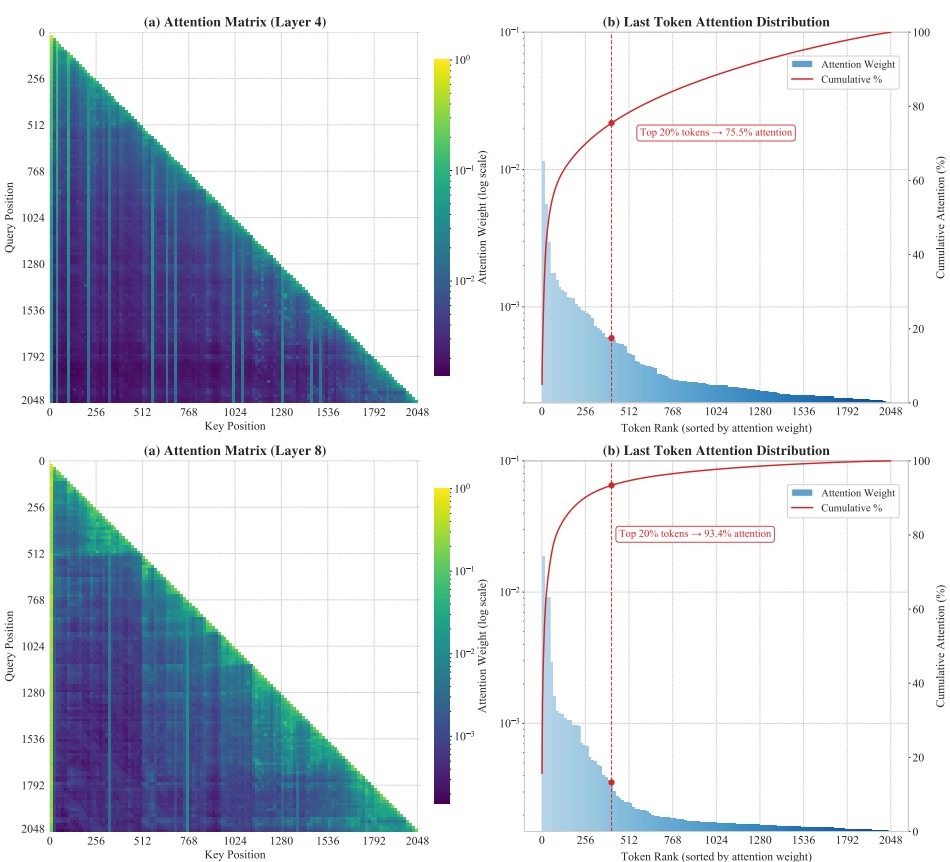

Figure 8: Attention map and last-token attention distribution visualization for Layers 4/8 (of 16) in OLMoE-7A1B.

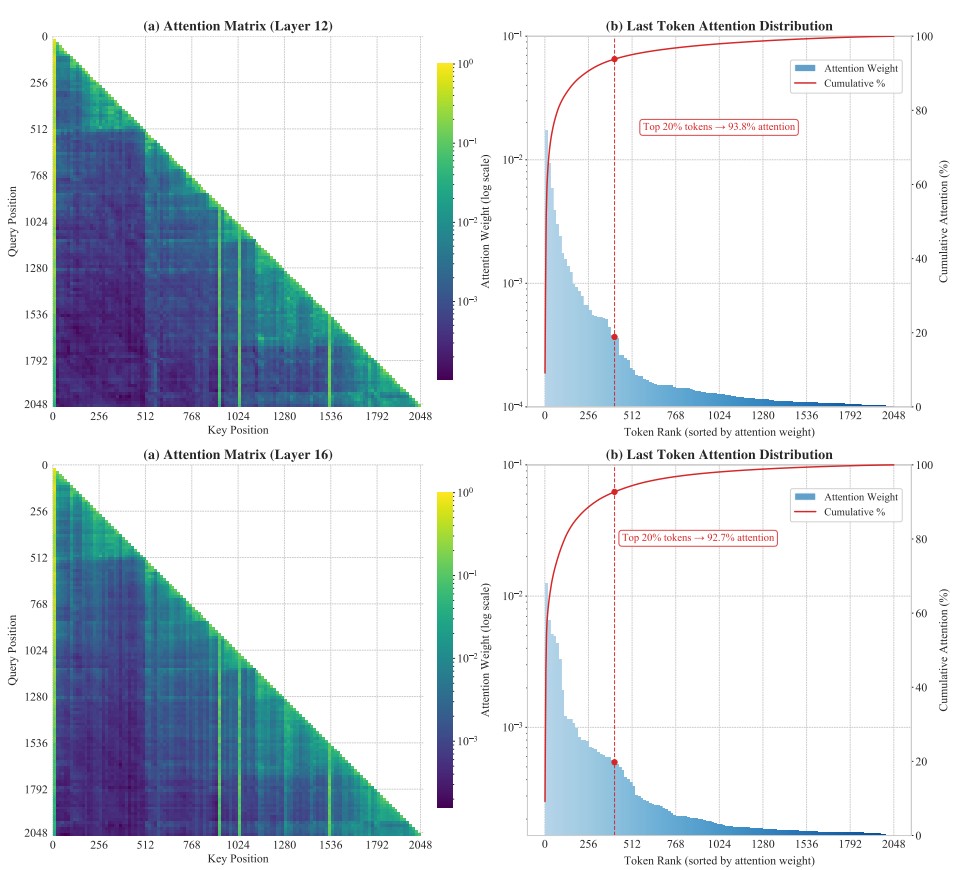

Figure 9: Attention map and last-token attention distribution visualization for Layers 12/16 (of 16) in OLMoE-7A1B.

# I   VISUAL ANALYSIS OF ROUTING DISTRIBUTIONS BEFORE AND AFTER EXPERT PRUNING

We compare the expert load across C4, Wikitext2, and Arc-challenge tasks. As clearly shown in the Figure 10 and Figure 11, after expert pruning, the routing patterns consistently exhibit minimal divergence, demonstrating that our bias updating strategy effectively compensates for expert removal without destabilizing the routing mechanism.

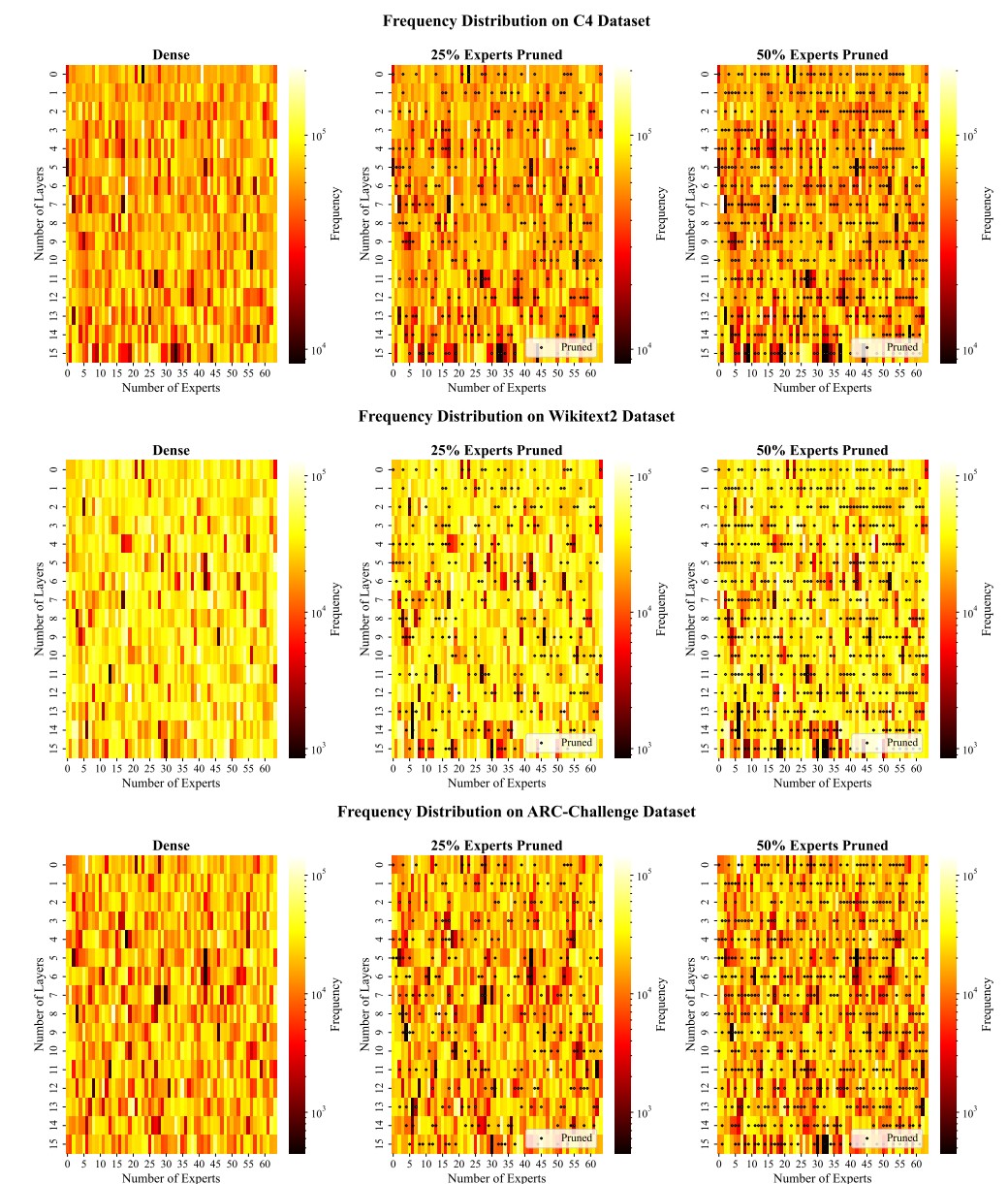

Figure 10: Expert routing frequency distribution of OLMoE-7A1B across C4, Wikitext2 and ARC-Challenge dataset.

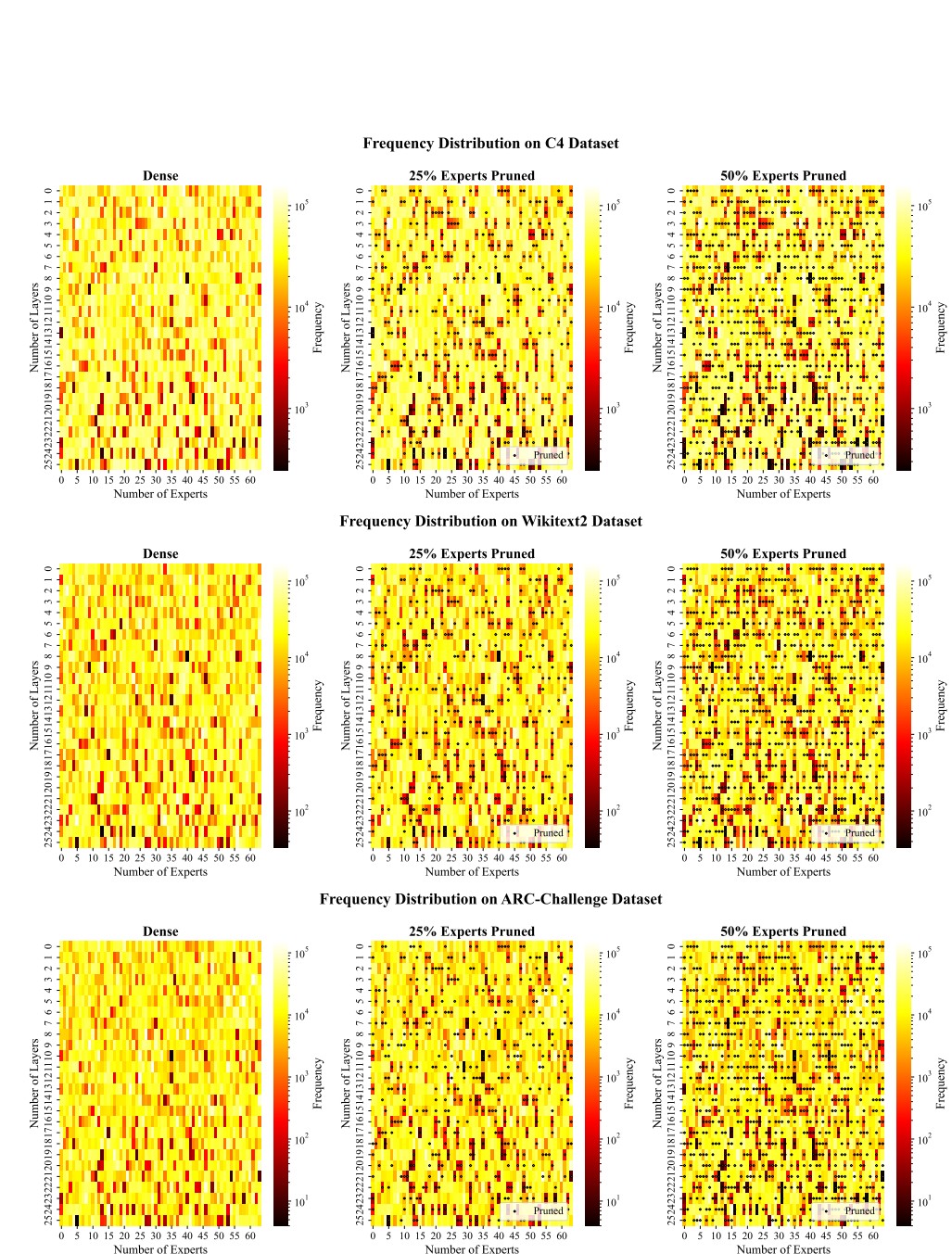

Figure 11: Expert routing frequency distribution of Moonlight-16A3B across C4, Wikitext2 and ARC-Challenge dataset.

