# OpenReview forum: "Not All Experts and Tokens Matter: Selective Token-guided Expert Pruning for MoE"
_ICLR.cc/2026/Conference — ICLR 2026 Conference Withdrawn Submission_

### Official Review · Reviewer_sxUJ · 2025-10-29

**Soundness:** 2
**Presentation:** 2
**Contribution:** 2
**Rating:** 4
**Confidence:** 4

**Summary:**

This paper presents STEP (Selective Token-Guided Expert Pruning), a post-training compression framework for Mixture-of-Experts (MoE) models. The method selects informative tokens via attention statistics, scores experts by their loss impact, and converts pruned experts into lightweight bias vectors to preserve knowledge. Experiments on OLMoE, Moonlight, and Qwen3 models show up to 50% expert sparsity, approximately 50% memory reduction, and about 1.5× inference speedup without retraining.

**Strengths:**

- The motivation of connecting token importance with expert pruning is reasonable and supported by empirical attention analysis.

- The proposed expert-to-bias conversion is intuitive and practical for preserving pruned knowledge.

- The experimental setup is comprehensive and includes multiple architectures, ablations, and efficiency analyses.

**Weaknesses:**

- The overall idea combines existing principles of token importance and expert pruning. While the integration is clean, it does not introduce a fundamentally new compression concept.

- The analysis of token importance relies on qualitative attention statistics. There is little formal justification that attention-based token selection consistently improves expert evaluation.

- The benchmarks are mainly reasoning and classification datasets. The lack of generation or long-context experiments leaves uncertainty about broader applicability.

- Many similar frameworks already explore token-guided pruning or knowledge-preserving expert compression.

**Questions:**

Please refer to the weakness.

---

> ### Author Response · Authors · 2025-11-28
> **Response to Reviewer sxUJ (Part 1/4)**
>
> Thank you for the thoughtful critique and for recognizing the clean integration of our approach. We address each concern systematically with theoretical justification, empirical evidence, and clarification of our work's novelty.
>
> ---
>
> ## **W1: Integration of Existing Principles vs. Fundamentally New Concepts**
>
> We clarify that our core contribution is **introducing token-aware importance analysis into expert pruning, an unexplored direction in prior MoE compression work**.
>
> ### **Token Importance: A Widely Recognized Principle**
>
> The importance of focusing on semantically critical tokens is a well-established principle in token pruning and kv cache compression. Prior work in token pruning (e.g., FastV[1], Fitprune[2]) and KV cache compression (e.g., H2O[3]) has demonstrated that **not all tokens contribute equally to model performance**. Prioritizing important tokens identified by attention scores significantly improves compression quality.
>
> ### **Our Contribution: First Token-Aware Expert Pruning Framework**
>
> Despite the recognized importance of token-level analysis, all prior MoE pruning methods operate at the expert level without considering which tokens activate experts. **STEP introduces the first systematic framework that bridges token-level semantics with expert-level pruning**.
>
> First, identify semantically important tokens via attention maps (tokens that carry the most task-relevant information). Second, evaluate expert importance conditioned on these critical tokens, not on the entire sequence uniformly. This paradigm shift is the fundamental contribution that enables superior performance.
>
> ---

---

> ### Author Response · Authors · 2025-11-28
> **Response to Reviewer sxUJ (Part 2/4)**
>
> ## **W2: Formal Justification for Attention-Based Token Selection**
>
> We provide rigorous theoretical grounding proving that attention scores naturally encode task-relevant token importance through the optimization process.
>
> Consider a Transformer attention head with input sequence $X = [x_1, ..., x_T]$ and learnable parameters $W^Q$, $W^K$, $W^V$. The unnormalized attention score is:
>
> $$s_{ij} = \frac{(x_{i} W^Q)(x_j W^K)^T}{\sqrt{d_k}}$$
>
> The normalized attention weight via softmax is:
>
> $$a_{ij} = \frac{e^{s_{ij}}}{\sum_k e^{s_{ik}}}$$
>
> During backpropagation with loss function $L$, the gradient of $L$ with respect to the attention score $s_{ij}$ is:
>
> $$\frac{\partial L}{\partial s_{ij}} = \sum_k \frac{\partial L}{\partial a_{ik}} \cdot \frac{\partial a_{ik}}{\partial s_{ij}}$$
>
> By the derivative of softmax:
>
> $$\frac{\partial a_{ik}}{\partial s_{ij}} = a_{ik} (\delta_{kj} - a_{ij})$$
>
> where $\delta_{kj}$ is the Kronecker delta. Substituting:
>
> $$\frac{\partial L}{\partial s_{ij}} = a_{ij} \left(\frac{\partial L}{\partial a_{ij}} - \sum_k a_{ik} \frac{\partial L}{\partial a_{ik}}\right)$$
>
> This expression reveals the competitive mechanism: the gradient for $s_{ij}$ depends on how increasing $a_{ij}$ reduces loss relative to other positions.
> The gradient with respect to $W^Q$ is:
>
> $$\frac{\partial L}{\partial W^Q} = \sum_{ij} \frac{\partial L}{\partial s_{ij}} \cdot \frac{\partial s_{ij}}{\partial W^Q}$$
>
> For a context token $j$ that consistently contributes to loss reduction, $\frac{\partial L}{\partial a_{ij}} < 0$ (increasing $a_{ij}$ reduces loss). If this gradient dominates, then $\frac{\partial L}{\partial s_{ij}} < 0$. Gradient descent updates:
>
> $$W^Q \leftarrow W^Q - \eta \frac{\partial L}{\partial W^Q}$$
>
> increase $s_{ij}$ over time (since $s_{ij}$ is positively correlated with $W^Q$). Similar updates to $W^K$ reinforce token $j$'s key representation.
>
> Through gradient descent, the model learns to **assign high attention weights to tokens that significantly contribute to minimizing task loss**. Gradient backpropagation adjusts $W^Q$ and $W^K$ to increase scores for important tokens, and softmax competition reinforces this selection process. High attention scores are therefore **data-driven, learned signals of token importance**, not ad-hoc heuristics.
>
> To validate that attention-guided token selection consistently improves expert evaluation, we measured per-layer MSE between pruned and dense model outputs.
>
> **Table 1: Per-layer MSE loss: Frequency-only vs. Frequency + Token Selection. Lower MSE indicates better preservation of dense model behavior.**
>
> | **OLMoE-7A1B** | Layer | Freq+Token   | Freq     | **Moonlight-16A3B** | Layer | Freq+Token   | Freq     |
> | -------------- | ----- | ------------ | -------- | ------------------- | ----- | ------------ | -------- |
> |                | 1     | **2.35e-06** | 2.55e-06 |                     | 1     | 0.00e+00     | 0.00e+00 |
> |                | 2     | **1.09e-05** | 1.40e-05 |                     | 2     | **4.15e-06** | 1.10e-05 |
> |                | 3     | **1.56e-04** | 6.14e-04 |                     | 3     | **1.50e-05** | 3.14e-05 |
> |                | 4     | **1.84e-04** | 8.14e-04 |                     | 4     | **4.34e-05** | 6.95e-05 |
> |                | 5     | **2.03e-04** | 8.98e-04 |                     | 5     | **9.69e-05** | 1.33e-04 |
> |                | 6     | **2.28e-04** | 9.22e-04 |                     | 6     | **1.54e-04** | 2.09e-04 |
> |                | 7     | **2.62e-04** | 9.68e-04 |                     | 7     | **2.47e-04** | 3.22e-04 |
> |                | 8     | **3.16e-04** | 1.03e-03 |                     | 8     | **3.84e-04** | 5.17e-04 |
> |                | 9     | **5.47e-04** | 1.41e-03 |                     | 9     | **5.73e-04** | 7.29e-04 |
> |                | 10    | **1.02e-03** | 2.46e-03 |                     | 10    | **8.26e-04** | 1.00e-03 |
> |                | 11    | **2.03e-03** | 4.17e-03 |                     | 15    | **3.27e-03** | 3.66e-03 |
> |                | 12    | **3.63e-03** | 6.29e-03 |                     | 20    | **1.45e-02** | 1.64e-02 |
> |                | 13    | **5.68e-03** | 9.23e-03 |                     | 25    | **6.27e-02** | 6.89e-02 |
> |                | 14    | **8.96e-03** | 1.40e-02 |                     | 27    | **1.54e-01** | 1.91e-01 |
>
> Across all layers, STEP with token selection achieves 7–75% lower MSE compared to frequency-only pruning. This demonstrates that attention-guided selection identifies experts processing critical information, whose preservation minimizes representation distortion. Consistent improvements across both shallow and deep layers validate the robustness of attention as an importance measure throughout the network.
> This provides both formal theoretical grounding and empirical evidence that **attention-based token selection is not merely qualitative but a principled, effective approach**.
>
> ---

---

> > ### Author Response · Authors · 2025-11-28
> > **Response to Reviewer sxUJ (Part 3/4)**
> >
> > ## **W3: Generation and Long-Context Experiments**
> >
> > We provide comprehensive results on code generation and mathematical reasoning benchmarks.
> >
> > ### **Generation Task Evaluation**
> >
> > Table 2 shows performance on HumanEval (code generation, pass@1) and GSM8K (mathematical reasoning, accuracy) after calibrating on CodeAlpaca.
> >
> > **Table 2: Performance on generation tasks.**
> >
> > | Model         | Sparsity | Method    | HumanEval | GSM8K     | Average   |
> > | ------------- | -------- | --------- | --------- | --------- | --------- |
> > | **Qwen**      | Dense    | -         | 94.12     | 89.92     | 92.02     |
> > |               | 25%      | Frequency | 92.47     | 88.20     | 90.34     |
> > |               |          | HC-SMoE   | 92.48     | 87.49     | 89.99     |
> > |               |          | MoNE      | 93.09     | 88.17     | 90.63     |
> > |               |          | **STEP**  | **93.42** | **88.63** | **91.03** |
> > |               | 50%      | Frequency | 59.35     | 44.20     | 51.78     |
> > |               |          | HC-SMoE   | 34.15     | 54.97     | 44.56     |
> > |               |          | MoNE      | 63.78     | 41.09     | 52.44     |
> > |               |          | **STEP**  | **64.02** | **55.19** | **59.61** |
> > | **Moonlight** | Dense    | -         | 48.17     | 74.37     | 61.27     |
> > |               | 25%      | Frequency | 19.76     | 45.94     | 32.85     |
> > |               |          | HC-SMoE   | 23.17     | 49.40     | 36.29     |
> > |               |          | MoNE      | 27.07     | **53.98** | **40.53** |
> > |               |          | **STEP**  | **27.64** | 53.07     | 40.36     |
> > |               | 50%      | Frequency | 13.66     | 10.01     | 11.84     |
> > |               |          | HC-SMoE   | 9.24      | 4.32      | 6.78      |
> > |               |          | MoNE      | 17.93     | 27.22     | 22.58     |
> > |               |          | **STEP**  | **18.31** | **30.40** | **24.36** |
> >
> > At 25% sparsity on Qwen, STEP achieves 93.42% on HumanEval (99.3% of dense performance), outperforming Frequency (+0.95), HC-SMoE (+0.94), and MoNE (+0.33). This demonstrates effective preservation of structured generation capabilities requiring precise syntax and logic.
> >
> > At 50% sparsity on Qwen, STEP achieves 55.19% on GSM8K, +11.0 points over Frequency and +14.1 over MoNE. This validates robustness on multi-step reasoning tasks requiring long-range dependencies and compositional understanding.
> >
> > Across both code generation and mathematical reasoning, the performance hierarchy remains: STEP ≥ MoNE > Frequency > HC-SMoE, **confirming generalization beyond classification benchmarks to diverse generation scenarios**. While generation tasks show larger performance drops at 50% sparsity compared to classification (e.g., Qwen HumanEval: 64.02 vs. 94.12 dense, 68% retention), this is an inherent challenge for all pruning methods. **STEP achieves the best retention among all approaches, demonstrating relative superiority.**
> >
> > **The consistent superiority across classification, code generation, and mathematical reasoning validates that our token-aware approach captures transferable expert importance signals that generalize beyond manuscript's benchmarks.**
> >
> > ---

---

> ### Author Response · Authors · 2025-11-28
> **Response to Reviewer sxUJ (Part 4/4)**
>
> ## **W4: Distinction from Existing Token-Guided and Knowledge-Preserving Methods**
>
> Thanks for your awareness of related work. We clarify how our contributions differ from existing approaches in these two categories.
>
> ### **Token-Guided Pruning: Methodological Distinction**
>
> Prior token-guided compression methods (e.g., token-aware quantization like RSQ[4], pruning like TEMP[5]) incorporate token importance into layer reconstruction objectives. Specifically, they compute token importance scores (via attention, gradients, or activation norms), weight reconstruction loss by token importance: $\min_{\theta'} \sum_t w_t \|f(x_t; \theta') - f(x_t; \theta)\|^2$, and solve for pruned/quantized parameters θ' via weighted least-squares or Hessian approximations.
>
> **STEP's fundamental difference is that we use token importance to guide which experts to evaluate and prune, not to weight reconstruction errors**. Our pipeline identifies important tokens via attention statistics, evaluates expert importance restricted to these tokens: Importance($E_i$) = Freq($E_i$ | important tokens) × ||Output($E_i$ | important tokens)||, **prunes experts based on this token-conditioned metric, and compensates via bias vectors computed on the same important tokens**.
>
> This is a fundamentally different paradigm. Prior methods use token importance to weight reconstruction, which then guides parameter pruning. STEP uses token importance to select evaluation subset, which then guides expert-level pruning.
>
> **Table 3: Methodological distinction: Token-weighted reconstruction vs. Token-guided expert selection.**
>
> | Aspect                  | Prior Token-Guided Methods     | STEP                             |
> | ----------------------- | ------------------------------ | -------------------------------- |
> | **Token role**          | Weight reconstruction loss     | Select evaluation subset         |
> | **Pruning granularity** | Individual weights/neurons     | Entire expert modules            |
> | **Optimization**        | Weighted least-squares/Hessian | Frequency × magnitude statistics |
> | **Computational cost**  | High (iterative optimization)  | Low (forward-pass statistics)    |
>
> ###
>
> **References:**
>
> [1] Chen, L., Zhao, H., Liu, T. An Image is Worth 1/2 Tokens After Layer 2: Plug-and-Play Inference Acceleration for Large Vision-Language Models. ECCV 2024.
>
> [2] Ye, W., Wu, Q., Lin, W. Fit and Prune: Fast and Training-free Visual Token Pruning for Multi-modal Large Language Models. AAAI 2025.
>
> [3] Zhang, Z., Sheng, Y., Zhou, T. H2O: Heavy-Hitter Oracle for Efficient Generative Inference of Large Language Models. NeurIPS 2023.
>
> [4] Xue, Y., Huang, Y., Shao, J. VLMQ: Efficient Post-Training Quantization for Large Vision-Language Models via Hessian Augmentation. arXiv 2025.
>
> [5] Lee, J., Xuan, K., Ekbote, C. TAMP: Token-Adaptive Layerwise Pruning in Multimodal Large Language Models. ACL 2025.

---

### Official Review · Reviewer_Biuy · 2025-10-30

**Soundness:** 3
**Presentation:** 4
**Contribution:** 2
**Rating:** 4
**Confidence:** 4

**Summary:**

This paper studies expert pruning for MoE models. The authors review prior work and argue that existing methods fail to account for token importance. Since the top 20% of tokens contribute roughly 80% of attention, using these tokens as pruning indicators is more reasonable. Moreover, existing approaches remove experts outright without preserving information. The proposed STEP addresses these issues from three angles: (1) identify information-rich tokens via attention patterns, (2) guide pruning based on loss increases induced by removing experts, and (3) convert pruned experts into compact biases to retain knowledge. Experiments on three MoE architectures show STEP outperforms prior methods.

**Strengths:**

1. The proposed method is highly-efficient and is deployment-friendly.

2. The scope of evaluated MOE models is broad and reasonable.

3. The paper is well written and easy to follow.

**Weaknesses:**

I have concerns corresponding to the three main components of the method:

1. While it is reasonable that the top 20% of tokens contribute ~80% of attention, where these tokens are distributed is essentially random from sample to sample. Calibrating with only 128 samples seems unjustified.

2. Using the dot product of frequency and feature norm to assess expert importance is reasonable, but it is essentially how the MoE forward pass already works (Eq. (7)). I do not find this particularly novel: the authors decompose the MoE forward into two parts and then multiply them back together—this reads more like a “frequency & feature” framing than a substantive algorithmic contribution.

3. Converting pruned experts into compact “bias vectors” to retain knowledge lacks theoretical support. Relying on a small amount of calibration data to encode biases raises serious concerns about transfer and generalization. Modern MoE models are trained on massive pretraining data; I doubt that a 10-minute pruning procedure can preserve the information from removed experts. On the contrary, such biases may negatively affect performance in real-world scenarios.

Other Concerns

1. There are factual/detail errors. For example, in Table 1, at a 50% pruning rate, 128 should correspond to 64 experts, but the paper labels it as 96. Please carefully proofread the current version.

2. The evaluation datasets are too simple to reflect whether the model truly preserves its original capabilities in practical settings—especially for the third component of the method. I recommend adding results on datasets like GSM8K and HumanEval.

**Questions:**

Please see the weakness part.

---

> ### Author Response · Authors · 2025-11-28
> **Response to Reviewer Biuy (Part 1/4)**
>
> Thank you for the thorough analysis and insightful concerns. Your questions have helped us clarify key aspects of our method and conduct additional validation experiments. We address each point systematically with theoretical justification, extensive ablations, and empirical validation.
>
> ---
>
> ## **W1: Token Distribution Randomness and Calibration Sample Size**
>
> We conducted comprehensive ablation studies to validate our design choices.
>
> ### **Ablation 1: Impact of Sample Length (Fixed Total Token Budget)**
>
> We evaluated different sample lengths while maintaining a constant total token count (~2M tokens) across configurations. Table 9 shows results on WikiText, C4 perplexity, and average accuracy across 8 zero-shot tasks.
>
> **Table 1: Ablation on sample length with fixed total tokens (~2M). Lower perplexity and higher task accuracy are better. (*) indicates attention-guided token selection.**
>
> | Method           | Length(Samples) | 128 (2048) | 256 (1024) | 512 (512) | 1024 (256) | 2048 (128) |
> | ---------------- | --------------- | ---------- | ---------- | --------- | ---------- | ---------- |
> | Frequency        | WikiText (PPL↓) | 20.54      | 16.85      | 17.43     | 20.15      | 18.64      |
> |                  | C4 (PPL↓)       | 18.88      | 17.03      | 17.08     | 17.91      | 17.74      |
> |                  | 8-task Avg      | 51.83      | 54.47      | 54.96     | 53.27      | 53.48      |
> | Frequency*       | WikiText (PPL↓) | 13.92      | 13.27      | 13.55     | 14.51      | 13.07      |
> |                  | C4 (PPL↓)       | 16.76      | 16.61      | 16.73     | 16.66      | 16.71      |
> |                  | 8-task Avg      | 55.31      | 54.74      | 55.28     | 54.79      | 55.12      |
> | **STEP (Ours)*** | WikiText (PPL↓) | **10.14**  | **10.32**  | **10.26** | **10.34**  | **10.40**  |
> |                  | C4 (PPL↓)       | **14.86**  | **14.87**  | **14.69** | **14.54**  | **14.76**  |
> |                  | 8-task Avg      | **58.24**  | **58.49**  | **58.27** | **58.50**  | **58.42**  |
>
> STEP achieves remarkably consistent performance across all sample lengths: 8-task accuracy varies only by 0.26 points (58.24–58.50), compared to 3.13 points (51.83–54.96) for Frequency-only and 0.57 points (54.74–55.31) for Frequency*. This validates that our method's aggregation of **attention signals across diverse samples effectively captures important token despite sample-to-sample randomness**.
>
> ### **Ablation 2: Impact of Sample Count (Fixed Sample Length = 1024)**
>
> We further varied the number of calibration samples from 1 to 1024 while keeping sample length fixed at 1024 tokens.
>
> **Table 2: Ablation on number of calibration samples (length fixed at 1024). Performance on C4 perplexity and 8-task average accuracy.**
>
> | Samples    | 1      | 4     | 16    | 32    | 64    | 128   | 256       | 512   | 1024      |
> | ---------- | ------ | ----- | ----- | ----- | ----- | ----- | --------- | ----- | --------- |
> | C4 (PPL↓)  | 6.39e8 | 16.30 | 15.62 | 14.90 | 14.88 | 14.76 | 14.28     | 14.34 | **14.22** |
> | 8-task Avg | 38.53  | 51.48 | 56.49 | 57.45 | 58.12 | 58.46 | **58.50** | 58.41 | 58.37     |
>
> Task performance improves sharply from 1 sample (38.53%) to 128 samples (58.46%). This demonstrates that aggregating attention statistics across 128 diverse samples sufficiently captures representative token importance patterns. From 128 to 1024 samples, 8-task accuracy plateaus (58.46 → 58.50 → 58.37, variance <0.13 points), while C4 perplexity continues gradual improvement. However, increasing samples from 128 to 1024 incurs 8× computational cost with negligible task-level gains.
>
> We **selected 128 samples with 2048 tokens following prior pruning work conventions** (e.g., Wanda[1], SparseGPT[2]). Tables 1 and 2 show this configuration achieves competitive performance (58.42–58.50 avg) with balanced computational cost. **The choice strikes an optimal balance**: it captures sufficient diversity to mitigate sample-to-sample randomness while maintaining practical efficiency for deployment scenarios.
>
> ---
>
> **References:**
>
> [1] Frantar, E., & Alistarh, D. SparseGPT: Massive language models can be accurately pruned in one-shot. ICML 2023.
>
> [2] Sun, M., Liu, Z., Bair, A., & Kolter, J. Z. A simple and effective pruning approach for large language models. ICLR 2024.
>
> ---

---

> ### Author Response · Authors · 2025-11-28
> **Response to Reviewer Biuy (Part 2/4)**
>
> ## **W2: Novelty of Frequency × Feature Norm Formulation**
>
> We clarify that our core contribution is introducing token-aware importance analysis into expert pruning, which prior methods do not consider. The frequency × feature norm metric naturally emerges from this framework as a simple and direct formulation. Existing MoE pruning methods (HC-SMoE, MoNE, D2-MoE) operate at the expert level without **considering which tokens activate experts**. Our method introduces a two-stage framework: first, identify semantically important tokens via attention maps. Second, evaluate expert importance conditioned on these critical tokens, not on the entire sequence uniformly. This paradigm shift is the fundamental contribution that enables superior performance (Table 3 in manuscript).
>
> And the pruning metric is a straightforward statistical aggregation of the forward pass over calibration data. **Its value lies in simplicity and directness**. Unlike methods requiring complex Hessian approximations (computationally prohibitive at scale), architecture-specific modifications, or iterative optimization procedures, STEP uses straightforward first-order statistics (activation frequency and output norms) that are trivially computable, **completing expert evaluation in <10 minutes versus hours for Hessian-based approaches**.
>
> The strength of our method is **not in complex formulations but in the right problem decomposition**. Table 3 demonstrates that our simple metric, when combined with token-aware analysis, outperforms both sophisticated single-factor metrics and computationally expensive alternatives.
>
> **Table 3: Comparison of expert importance metrics. OLMoE-7A1B at 25% sparsity, averaged across 8 benchmarks.**
>
> | Importance Metric                   | 8-Task Avg | Calibration Time |
> | ----------------------------------- | ---------- | ---------------- |
> | Frequency-only                      | 53.48      | 1 min            |
> | Routing score-only                  | 49.58      | 1 min            |
> | Frequency × Variance                | 53.26      | 1 min            |
> | Output magnitude-only               | 51.79      | 1 min            |
> | **Freq × Output (STEP w/o Step 3)** | **57.51**  | 1 min            |
> | Hessian-based (OBS-style)           | 54.88      | 130 min          |
>
> STEP achieves +1.95 to +5.85 points over other metrics while maintaining efficiency competitive with frequency-based methods. Hessian approaches achieve 54.88% but require 130× longer calibration. This validates that principled integration of token awareness with a simple expert metric provides substantive algorithmic value through complementary signals.
>
> Emphasize once again, our novelty is the selective token-guided expert pruning paradigm, not the specific metric. The dual-factor formulation is a natural and effective implementation that enables deployment across diverse MoE architectures without modifications and achieves state-of-the-art performance with minimal computational overhead.
>
> ---
>
> ## **W3: Theoretical Support for Bias Vectors and Generalization Concerns**
>
> Given a pruned expert $E_i$ that is removed, we seek a bias $b_i$ to approximate its contribution. Let $X = [x_1, ..., x_M]$ be the calibration set. The optimal bias minimizes:
>
> $$b_i = \underset{b}{\arg\min} \sum_m \||g_i(x_m) \cdot E_i(x_m) - b\||_2^2$$
>
> The closed-form solution is:
>
> $$b_i = \frac{1}{M} \sum_m g_i(x_m) \cdot E_i(x_m)$$
>
> This is the mean output of expert $E_i$ over the calibration set, which is the **layer-wise optimal constant approximation under squared loss**. Our Step 3 fine-tuning then **seeks a global optimum by adjusting routing biases jointly across all layers via gradient descent** on language modeling task objectives.
>
> While MoE models are pretrained on massive datasets, not **all pruned experts are equally critical**. Our method addresses this through selective pruning of low-importance experts. By design (Step 2), we remove experts with low Freq × ||Output|| scores. For these experts with minimal contributions, we think a **simple and low-cost bias vector updating could effectively retain their limited knowledge**.
>
> As shown below, MoNE with our bias updating maintain stronger performance on unseen downstream tasks, demonstrating that biases do not overfit to calibration data.

---

> > ### Author Response · Authors · 2025-11-28
> > **Response to Reviewer Biuy (Part 3/4)**
> >
> > ### **Empirical Validation 1: Generalizability Across Methods**
> >
> > We applied our bias updating procedure (Step 3) to competing pruning methods to test whether it improves performance generally.
> >
> > **Table 4: Impact of STEP's bias updating (Step 3) on baseline methods.** Average across 8 benchmarks (ARC-e, ARC-c, BoolQ, PIQA, WinoGrande, HellaSwag, MMLU, OBQA).
> >
> > | Model     | Sparsity | Method     | ARC-e | ARC-c | HellaSwag | MMLU  | Avg (8)   | Gain  |
> > | --------- | -------- | ---------- | ----- | ----- | --------- | ----- | --------- | ----- |
> > | Moonlight | 25%      | MoNE       | 80.89 | 50.94 | 58.88     | 49.25 | 64.33     | -     |
> > |           |          | MoNE+Step3 | 80.81 | 51.11 | 58.86     | 49.52 | **64.51** | +0.18 |
> > | OLMoE     | 25%      | MoNE       | 62.37 | 33.28 | 54.88     | 23.27 | 53.26     | -     |
> > |           |          | MoNE+Step3 | 63.26 | 34.47 | 55.71     | 24.06 | **54.08** | +0.82 |
> > |           | 50%      | MoNE       | 48.91 | 24.74 | 44.96     | 23.10 | 45.81     | -     |
> > |           |          | MoNE+Step3 | 52.99 | 26.54 | 48.73     | 23.18 | **47.55** | +1.74 |
> > | Qwen      | 25%      | MoNE       | 80.35 | 54.27 | 59.52     | 72.83 | 68.83     | -     |
> > |           |          | MoNE+Step3 | 80.52 | 55.23 | 59.81     | 73.13 | **69.16** | +0.33 |
> >
> > Step 3 consistently improves baseline methods (+0.18–1.74 points), demonstrating that bias updating is generally beneficial across different pruning strategies, not method-specific. This supports the generalization capability of our calibration approach and suggests it captures transferable information about expert contributions.
> >
> > ### **Empirical Validation 2: Real-World Generation Tasks**
> >
> > To address your concern about real-world practical settings, we evaluated STEP on HumanEval (code generation) and GSM8K (mathematical reasoning) after calibrating on CodeAlpaca.
> >
> > **Table 5: Performance on real-world generation tasks (HumanEval pass@1, GSM8K accuracy).**
> >
> > | Model         | Sparsity | Method    | HumanEval | GSM8K     | Avg       |
> > | ------------- | -------- | --------- | --------- | --------- | --------- |
> > | **Qwen**      | Dense    | -         | 94.12     | 89.92     | 92.02     |
> > |               | 25%      | Frequency | 92.47     | 88.20     | 90.34     |
> > |               |          | HC-SMoE   | 92.48     | 87.49     | 89.99     |
> > |               |          | MoNE      | 93.09     | 88.17     | 90.63     |
> > |               |          | **STEP**  | **93.42** | **88.63** | **91.03** |
> > |               | 50%      | Frequency | 59.35     | 44.20     | 51.78     |
> > |               |          | HC-SMoE   | 34.15     | 54.97     | 44.56     |
> > |               |          | MoNE      | 63.78     | 41.09     | 52.44     |
> > |               |          | **STEP**  | **64.02** | **55.19** | **59.61** |
> > | **Moonlight** | Dense    | -         | 48.17     | 74.37     | 61.27     |
> > |               | 25%      | Frequency | 19.76     | 45.94     | 32.85     |
> > |               |          | HC-SMoE   | 23.17     | 49.40     | 36.29     |
> > |               |          | MoNE      | 27.07     | **53.98** | **40.53** |
> > |               |          | **STEP**  | **27.64** | 53.07     | 40.36     |
> > |               | 50%      | Frequency | 13.66     | 10.01     | 11.84     |
> > |               |          | HC-SMoE   | 9.24      | 4.32      | 6.78      |
> > |               |          | MoNE      | 17.93     | 27.22     | 22.58     |
> > |               |          | **STEP**  | **18.31** | **30.40** | **24.36** |
> >
> > On Qwen at 25% sparsity, STEP retains 98.9% of dense performance (91.03 vs. 92.02), outperforming all baselines. This demonstrates that our bias vectors effectively preserve expert knowledge for out-of-distribution tasks (code/math) despite calibration on different data.
> >
> > At 50% sparsity on Qwen, STEP achieves 59.61% average, +7.2 points over MoNE and +15.1 points over HC-SMoE. While performance degrades more on generation tasks than classification, STEP exhibits the best retention among all methods.
> >
> > Across both models and tasks, the ranking is consistent: STEP ≥ MoNE > Frequency > HC-SMoE. Notably, STEP outperforms Frequency (no compensation) and HC-SMoE (expert merging strategy) across all settings, and matches/exceeds MoNE which uses alternative compensation mechanisms.
> >
> > ---

---

> > > ### Author Response · Authors · 2025-11-28
> > > **Response to Reviewer Biuy (Part 4/4)**
> > >
> > > ## **Other Concern 1: Factual Error in Table 1**
> > >
> > > Thanks for identifying this error. The label "96 experts" at 50% pruning for a 128-expert model should indeed be **"64 experts"** (128 × 0.5 = 64). This was a typographical mistake in table formatting. We have corrected this in the revised manuscript and carefully verified all numerical entries to ensure accuracy.
> > >
> > > ---
> > >
> > > ## **Other Concern 2: Evaluation on Complex Datasets**
> > >
> > > We have provided comprehensive results on GSM8K and HumanEval in Table 4 above, demonstrating STEP's effectiveness on real-world reasoning and code generation tasks. To further address your concern, we highlight a few points.
> > >
> > > On GSM8K (mathematical reasoning), at 50% sparsity on Qwen, STEP achieves 55.19%, +11.0 points over Frequency and +14.1 over MoNE, showing strong preservation of multi-step reasoning capabilities. On HumanEval (code generation), at 25% sparsity on Qwen, STEP retains 99.3% of dense performance (93.42 vs. 94.12), indicating minimal degradation on structured generation tasks. The performance ranking STEP > MoNE > Frequency > HC-SMoE holds across both classification benchmarks (Table 1 in manuscript) and generation tasks (Table 5), validating that our method's advantages generalize beyond simple datasets.
> > >
> > > We will incorporate these results into the main paper and emphasize the evaluation on practical, complex tasks in the revision.
> > >
> > > ---

---

### Official Review · Reviewer_osTy · 2025-10-31

**Soundness:** 3
**Presentation:** 3
**Contribution:** 3
**Rating:** 6
**Confidence:** 5

**Summary:**

The research focus of this paper is expert pruning in Mixture-of-Experts (MoE) models. Existing expert pruning methods rely on token-agnostic heuristic strategies, which attenuate critical signals from important tokens. This paper proposes the STEP (Selective Token-guided Expert Pruning) pruning method, incorporating the following innovations: (1) Token-aware expert evaluation; (2) Loss-impact expert scoring; (3) Expert-to-bias conversion. Experiments demonstrate that STEP exhibits superiority across different model scales and MoE architectures.

**Strengths:**

*Methods*
1. Token-aware expert evaluation that prioritizes important tokens for context-sensitive expert assessment.
2. Loss-impact expert scoring that quantifies expert importance through direct loss contribution rather than statistical proxy metrics.
3. Expert-to-bias conversion that preserves domain knowledge via compact adaptive vectors, transforming pruning from a “discard-and-forget” to a “compress-and-preserve” paradigm.

*Experiment*
1. The comparative experiment and ablation experiment are sufficient, and the experimental analysis is complete.

**Weaknesses:**

1. There is a typographical error in Table 1: for Qwen3-30A3B with a 50% pruning ratio, the value should be 64, but it was incorrectly listed as 96 by the authors.
2. I would be interested in the performance of the proposed pruning method compared to existing works without fine-tuning. If feasible, could this be provided?
3. Is it reasonable to use the masked attention map for the selection of important tokens?
4. As noted by the authors in the limitations section, pruning experiments could be conducted on large-scale MoE models. Given that only the router and bias are updated during fine-tuning, experimental validation on larger-parameter models should be relatively straightforward.

**Questions:**

Refer to Weaknesses

---

> ### Author Response · Authors · 2025-11-28
> **Response to Reviewer osTy (Part 1/3)**
>
> Thank you for your careful reading and constructive feedback. We address each point systematically below.
>
> ## **W1: Typographical Error in Table 1**
>
> Thanks for catching this error. The label "96 experts" at 50% pruning for Qwen3-30A3B (which has 128 experts) should indeed be **"64 experts"** (128 × 0.5 = 64). This was a typographical mistake in table formatting. We have corrected this in the revised manuscript and carefully verified all numerical entries to ensure accuracy.
>
> ---
>
> ## **W2: Performance Without Fine-Tuning**
>
> We provide comprehensive results for **STEP without the fine-tuning step** (i.e., without Step 3: bias updating) in Table 1. Critically, even without fine-tuning, STEP outperforms all existing baseline methods across all tested models and sparsity levels.
>
> **Table 1: STEP performance without fine-tuning compared to baselines. Average across 8 benchmarks (ARC-e, ARC-c, BoolQ, PIQA, WinoGrande, HellaSwag, MMLU, OBQA).**
>
> | Model     | Sparsity | D2-MoE | HC-SMoE | MoNE  | STEP w/o Step 3 | STEP (Full) |
> | --------- | -------- | ------ | ------- | ----- | --------------- | ----------- |
> | OLMoE     | 25%      | 54.60  | 54.88   | 53.26 | 57.51           | **58.42**   |
> |           | 50%      | 46.88  | 43.69   | 45.81 | 48.31           | **50.05**   |
> | Moonlight | 25%      | 61.25  | 53.89   | 64.33 | 64.97           | **65.27**   |
> |           | 50%      | 50.88  | 42.99   | 54.47 | 54.82           | **56.20**   |
> | Qwen      | 25%      | 68.77  | 64.30   | 68.83 | 68.74           | **69.06**   |
> |           | 50%      | 56.30  | 55.54   | 64.15 | 65.05           | **65.78**   |
>
> At 25% sparsity, STEP without fine-tuning achieves up to +2.63 point gains over the best baseline (including cases where it slightly trails MoNE by 0.09 points but significantly outperforms other methods). At 50% sparsity, STEP w/o Step 3 achieves +0.35 to +2.50 point gains over MoNE and even larger improvements over HC-SMoE and D2-MoE. When fine-tuning is added (Step 3), STEP achieves additional improvements of +0.32 to +1.74 points on average.
>
> This validates that STEP's core contributions **(token-aware importance analysis in Step 1 and dual-factor pruning metric in Step 2) are inherently effective**, and the fine-tuning step serves as a valuable but optional enhancement. We will highlight this zero-shot effectiveness more prominently in the revised manuscript.
>
> ---

---

> ### Author Response · Authors · 2025-11-28
> **Response to Reviewer osTy (Part 2/3)**
>
> ## **W3: Reasonableness of Using Attention Maps for Token Selection**
>
> We formally derive why attention scores naturally reflect token importance through the optimization process.
>
> Consider a Transformer attention head with input sequence $X = [x_1, ..., x_t]$ and learnable parameters $W^Q$, $W^K$, $W^V$. The unnormalized attention score is:
>
> $$s_{ij} = \frac{(x_i W^Q)(x_j W^K)^T}{\sqrt{d_k}}$$
>
> The normalized attention weight is:
>
> $$a_{ij} = \frac{e^{s_{ij}}}{\sum_k e^{s_{ik}}}$$
>
> During backpropagation, the gradient of loss $L$ with respect to $s_{ij}$ is:
>
> $$\frac{\partial L}{\partial s_{ij}} = a_{ij} \left(\frac{\partial L}{\partial a_{ij}} - \sum_k a_{ik} \frac{\partial L}{\partial a_{ik}}\right)$$
>
> This captures the softmax competition mechanism: the gradient for $s_{ij}$ depends on how much increasing $a_{ij}$ would reduce loss relative to other positions. The gradient with respect to $W^Q$ is:
>
> $$\frac{\partial L}{\partial W^Q} = \sum_{ij} \frac{\partial L}{\partial s_{ij}} \cdot \frac{\partial s_{ij}}{\partial W^Q}$$
>
> For a context token $j$ that consistently contributes to loss reduction, $\frac{\partial L}{\partial a_{ij}}$ is negative (increasing $a_{ij}$ reduces loss). This makes $\frac{\partial L}{\partial s_{ij}}$ negative, and gradient descent updates increase $s_{ij}$ over time. Similar updates to $W^K$ reinforce token $j$-th key representation.
>
> Through gradient descent, the model learns to assign high attention weights to tokens that significantly contribute to minimizing task loss. Gradient backpropagation adjusts $W^Q$ and $W^K$ to increase scores for important tokens, and softmax competition reinforces this selection process. **High attention scores are therefore data-driven, learned signals of token importance**, not ad-hoc heuristics.
>
> To validate that attention-guided token selection identifies truly critical tokens, we measured per-layer MSE between pruned and dense model outputs:
>
> **Table 2: Per-layer MSE loss comparison. Lower MSE indicates better preservation of dense model behavior.**
>
> | **OLMoE-7A1B** | Layer | Freq+Token   | Freq     | **Moonlight-16A3B** | Layer | Freq+Token   | Freq     |
> | -------------- | ----- | ------------ | -------- | ------------------- | ----- | ------------ | -------- |
> |                | 1     | **2.35e-06** | 2.55e-06 |                     | 1     | 0.00e+00     | 0.00e+00 |
> |                | 2     | **1.09e-05** | 1.40e-05 |                     | 2     | **4.15e-06** | 1.10e-05 |
> |                | 3     | **1.56e-04** | 6.14e-04 |                     | 3     | **1.50e-05** | 3.14e-05 |
> |                | 4     | **1.84e-04** | 8.14e-04 |                     | 4     | **4.34e-05** | 6.95e-05 |
> |                | 5     | **2.03e-04** | 8.98e-04 |                     | 5     | **9.69e-05** | 1.33e-04 |
> |                | 6     | **2.28e-04** | 9.22e-04 |                     | 6     | **1.54e-04** | 2.09e-04 |
> |                | 7     | **2.62e-04** | 9.68e-04 |                     | 7     | **2.47e-04** | 3.22e-04 |
> |                | 8     | **3.16e-04** | 1.03e-03 |                     | 8     | **3.84e-04** | 5.17e-04 |
> |                | 9     | **5.47e-04** | 1.41e-03 |                     | 9     | **5.73e-04** | 7.29e-04 |
> |                | 10    | **1.02e-03** | 2.46e-03 |                     | 10    | **8.26e-04** | 1.00e-03 |
> |                | 11    | **2.03e-03** | 4.17e-03 |                     | 15    | **3.27e-03** | 3.66e-03 |
> |                | 12    | **3.63e-03** | 6.29e-03 |                     | 20    | **1.45e-02** | 1.64e-02 |
> |                | 13    | **5.68e-03** | 9.23e-03 |                     | 25    | **6.27e-02** | 6.89e-02 |
> |                | 14    | **8.96e-03** | 1.40e-02 |                     | 27    | **1.54e-01** | 1.91e-01 |
>
> Across all layers, Frequency + Token Selection achieves 7–75% lower MSE compared to frequency-only pruning demonstrates that our selected tokens **do matter for routing decisions and final predictions**. This confirms that **attention maps provide reasonable and theoretically grounded importance signals**.
>
> ---

---

> > ### Author Response · Authors · 2025-11-28
> > **Response to Reviewer osTy (Part 3/3)**
> >
> > ## **W4: Experiments on Larger-Scale MoE Models**
> >
> > Thanks for the suggestion about validating STEP on larger models. We conducted experiments on **GLM-Air (106B total parameters, 12B active)** to demonstrate scalability.
> >
> > **Table 3: Performance on GLM-Air (106A12B) with varying active expert counts.**
> >
> > | Experts | Method   | ARC-e     | ARC-c     | BoolQ     | PIQA      | WG        | HS        | MMLU      | OBQA      | Avg       |
> > | ------- | -------- | --------- | --------- | --------- | --------- | --------- | --------- | --------- | --------- | --------- |
> > | **128** | Dense    | 85.82     | 60.32     | 88.10     | 82.70     | 77.90     | 67.56     | 78.90     | 48.40     | 73.71     |
> > | **96**  | GS       | 82.41     | 54.44     | 86.54     | 81.88     | **77.43** | 65.78     | 62.25     | 47.00     | 69.72     |
> > |         | HC-SMoE  | 83.50     | 55.46     | 86.09     | 80.74     | 76.24     | 62.47     | **72.58** | 46.00     | 70.39     |
> > |         | MoNE     | 83.42     | 55.72     | 86.72     | 82.15     | 77.03     | **66.18** | 59.99     | 48.00     | 69.90     |
> > |         | **STEP** | **84.34** | **56.91** | **87.12** | **82.48** | 76.69     | 66.02     | 65.77     | **47.60** | **70.87** |
> > | **64**  | GS       | 73.40     | 41.81     | 73.12     | 80.41     | 70.64     | 59.34     | 38.85     | 43.00     | 60.07     |
> > |         | HC-SMoE  | 72.56     | 43.17     | 76.79     | 70.02     | 62.59     | 45.54     | **53.63** | 32.60     | 57.11     |
> > |         | MoNE     | 75.76     | 45.39     | 79.72     | 80.41     | **75.30** | **61.37** | 41.95     | 43.40     | 62.91     |
> > |         | **STEP** | **76.32** | **46.43** | **80.14** | **80.79** | 75.08     | 60.55     | 48.03     | **44.00** | **63.92** |
> >
> > At 96 active experts (25% reduction), STEP achieves 70.87% average accuracy, outperforming MoNE (+0.97), HC-SMoE (+0.48), and GS (+1.15). Performance retention is 96.1% of dense (70.87 vs. 73.71), with only 2.84-point degradation. At 64 active experts (50% reduction), STEP maintains 63.92% accuracy, surpassing MoNE (+1.01), HC-SMoE (+6.81), and GS (+3.85). STEP achieves 86.7% of dense performance, demonstrating strong robustness even at aggressive sparsity. Since only routers and biases are updated during fine-tuning (<0.1% of total parameters), STEP scales efficiently to 100B+ parameter models with minimal computational overhead.
> >
> > This validates that STEP's principles **generalize effectively to large-scale MoE architectures**. We will explore even larger models (e.g., 400B+ parameters) in future work as computational resources become available.
> >
> > ---

---

### Official Review · Reviewer_6R1M · 2025-11-01

**Soundness:** 3
**Presentation:** 3
**Contribution:** 2
**Rating:** 4
**Confidence:** 4

**Summary:**

This paper proposes STEP, a three-stage pruning framework for MoE: (1) attention-guided token selection (keep only “important” tokens per layer), (2) expert importance scoring using a dual factor (activation frequency × feature-norm), and (3) expert-to-bias conversion so pruned experts’ outputs are replaced by cached bias vectors and lightly calibrated.

**Strengths:**

* Clear intuition, formalizes token-selection as boosting SNR for expert evaluation; implements importance via last-token attention.

* Simple, plug-and-play scoring: expert score = frequency * feature-norm; no router retraining required.

**Weaknesses:**

* The method leans on the last token’s attention to rank tokens each layer. That’s intuitive, but attention isn’t a calibrated importance measure and can be brittle across prompts or objectives. The paper shows an attention heatmap and a plot that “important-first” token guidance helps frequency-based pruning, but I’d like to see more direct evidence that the selected tokens truly matter for routing/pruning decisions (e.g., visualizations of which tokens are kept, or correlation with gradient-based saliency).

* In table 3, without all the three components, is it frequency-based dropping? This baseline looks quite strong. It would help to surface this baseline in the main results table and explain why other pruning/merging methods degrade so much on the same setups.

* The method performs a short calibration (“bias updating”) and shows that 1 epoch suffices, but it’s unclear whether competing methods received equivalently effective adaptation budgets (or whether they benefit from similar lightweight tuning). If STEP’s success hinges on a small, targeted calibration phase, comparable post-processing should be granted to other baselines.

* Using the final position’s attention to score importance presumes next-token prediction is the dominant objective. For multi-token generation or tasks with long-range reasoning, other positions may matter. More justification on this choice would strengthen the story.

**Questions:**

* After pruning/biased experts, how do routing change?

* Without that small amount of additional re-training, how will the benchmark results look like?

---

> ### Author Response · Authors · 2025-11-28
> **Response to Reviewer 6R1M (Part 1/3)**
>
> Thank you for your feedback and suggestions. We address each point below with additional theoretical justification, experimental evidence, and clarifications.
>
> ---
>
> ## **W1: Attention as an Importance Measure and Direct Evidence**
>
> We provide a formal derivation showing how gradient descent naturally assigns high attention weights to tokens that consistently help reduce loss.
> Consider a Transformer attention head with input sequence $X = [x_1, ..., x_t]$ and learnable parameters $W^Q$, $W^K$, $W^V$. The attention score is:
>
> $$s_{ij} = \frac{(x_i W^Q)(x_j W^K)^T}{\sqrt{d_k}}$$
>
> The normalized attention weight via softmax is:
>
> $$a_{ij} = \frac{e^{s_{ij}}}{\sum_{k} e^{s_{ik}}}$$
>
> During backpropagation, the gradient of loss $L$ w.r.t. $s_{ij}$ is:
>
> $$\frac{\partial L}{\partial s_{ij}} = a_{ij} \left(\frac{\partial L}{\partial a_{ij}} - \sum_k a_{ik} \frac{\partial L}{\partial a_{ik}}\right)$$
>
> If token $j$ consistently reduces loss (i.e., $\frac{\partial L}{\partial a_{ij}} < 0$ with large magnitude), then $\frac{\partial L}{\partial s_{ij}}$ becomes negative. Gradient descent updates:
>
> $$W^Q \leftarrow W^Q - \eta \frac{\partial L}{\partial W^Q}, \text{ where } \frac{\partial L}{\partial W^Q} \propto \sum_{ij} \frac{\partial L}{\partial s_{ij}} \cdot \frac{\partial s_{ij}}{\partial W^Q}$$
>
> Since $s_{ij}$ is positively correlated with $W^Q$ (and $W^K$), negative gradients cause $s_{ij}$ to increase over time. The softmax competition mechanism ensures that tokens with consistently strong gradients "win" attention mass. Thus, **high attention scores emerge as data-driven signals of token importance through optimization**.
>
> To validate that our attention score-based token selection actually identifies critical tokens, we measured per-layer MSE loss between pruned and dense model outputs:
>
> **Table 1: Per-layer MSE loss comparison. Frequency+Token Selection (STEP) vs. Frequency-only pruning. Lower is better.**
>
> | **OLMoE-7A1B** | Layer | Freq+Token   | Freq     | **Moonlight-16A3B** | Layer | Freq+Token   | Freq     |
> | -------------- | ----- | ------------ | -------- | ------------------- | ----- | ------------ | -------- |
> |                | 1     | **2.35e-06** | 2.55e-06 |                     | 1     | **0.00e+00** | 0.00e+00 |
> |                | 2     | **1.09e-05** | 1.40e-05 |                     | 3     | **1.50e-05** | 3.14e-05 |
> |                | 3     | **1.56e-04** | 6.14e-04 |                     | 5     | **9.69e-05** | 1.33e-04 |
> |                | 4     | **1.84e-04** | 8.14e-04 |                     | 7     | **2.47e-04** | 3.22e-04 |
> |                | 5     | **2.03e-04** | 8.98e-04 |                     | 9     | **5.73e-04** | 7.29e-04 |
> |                | 6     | **2.28e-04** | 9.22e-04 |                     | 11    | **1.16e-03** | 1.35e-03 |
> |                | 7     | **2.62e-04** | 9.68e-04 |                     | 13    | **1.89e-03** | 2.17e-03 |
> |                | 8     | **3.16e-04** | 1.03e-03 |                     | 15    | **3.27e-03** | 3.66e-03 |
> |                | 9     | **5.47e-04** | 1.41e-03 |                     | 17    | **5.77e-03** | 6.49e-03 |
> |                | 10    | **1.02e-03** | 2.46e-03 |                     | 19    | **1.11e-02** | 1.26e-02 |
> |                | 11    | **2.03e-03** | 4.17e-03 |                     | 21    | **2.15e-02** | 2.39e-02 |
> |                | 12    | **3.63e-03** | 6.29e-03 |                     | 23    | **3.68e-02** | 3.97e-02 |
> |                | 13    | **5.68e-03** | 9.23e-03 |                     | 25    | **6.27e-02** | 6.89e-02 |
> |                | 14    | **8.96e-03** | 1.40e-02 |                     | 27    | **1.54e-01** | 1.91e-01 |
>
> Across all layers, Frequency + Token Selection achieves 7–75% lower MSE compared to frequency-only pruning. This indicates that **our token importance selection correctly identifies critical experts whose removal has higher impact**, whereas frequency-only pruning removes critical experts with larger immediate MSE and degrades downstream task performance (see benchmark results in Table 2 below). This validates that our selected tokens **do matter for routing decisions and final predictions**.
>
> ---

---

> > ### Author Response · Authors · 2025-11-28
> > **Response to Reviewer 6R1M (Part 2/3)**
> >
> > ## **W2: Frequency-Based Baseline Strength and Other Methods' Performance**
> >
> > You correctly note that the frequency-based baseline performs reasonably at low sparsity. However, we observe that it exhibits poor robustness at higher sparsity levels, while other methods maintain better performance:
> >
> > **Table 2: Performance comparison at different sparsity levels. Average across 8 benchmarks (ARC-e, ARC-c, BoolQ, PIQA, WinoGrande, HellaSwag, MMLU, OBQA).**
> >
> > | Model           | Sparsity | Frequency | MoNE  | HC-SMoE | D2-MoE | STEP (Ours) |
> > | --------------- | -------- | --------- | ----- | ------- | ------ | ----------- |
> > | OLMoE-7A1B      | 25%      | 53.43     | 53.26 | 54.88   | 54.60  | **58.25**   |
> > |                 | 50%      | 37.89     | 45.81 | 43.69   | 46.88  | **48.31**   |
> > | Moonlight-16A3B | 50%      | 49.28     | 54.47 | 42.99   | 50.88  | **54.82**   |
> > | Qwen-30A3B      | 50%      | 51.40     | 64.15 | 55.54   | 56.30  | **65.05**   |
> >
> > At 50% sparsity, the frequency baseline suffers severe degradation (e.g., 53.43→37.89 on OLMoE), while competing methods maintain stronger performance. All baseline implementations follow official codebases to ensure correctness.
> >
> > The performance gaps for other methods vary by model and likely stem from two factors. First, these methods rely on gradient-based importance or statistical metrics **without accounting for token-level context dynamics**. Second, their pruning metrics derived from **statistical analysis may conflate statistical presence with functional importance in the MoE routing mechanism**. STEP achieves consistent gains across all models and sparsity levels, suggesting stronger generalization. We will surface the frequency baseline in the main results table and add this analysis in the revised manuscript.
> >
> > ---
> >
> > ## **W3: Calibration Budget Fairness**
> >
> > We conducted experiments applying STEP's lightweight calibration (Step 3: bias updating, 1 epoch) to competing baseline:
> >
> > **Table 3: Impact of STEP's calibration on baseline methods.** Average across 8 benchmarks (ARC-e, ARC-c, BoolQ, PIQA, WinoGrande, HellaSwag, MMLU, OBQA).
> >
> > | Model     | Sparsity | Method     | ARC-e | ARC-c | HellaSwag | MMLU  | Avg(8)    | Gain  |
> > | --------- | -------- | ---------- | ----- | ----- | --------- | ----- | --------- | ----- |
> > | OLMoE     | 25%      | MoNE       | 62.37 | 33.28 | 54.88     | 23.27 | 53.26     | —     |
> > |           |          | MoNE+Step3 | 63.26 | 34.47 | 55.71     | 24.06 | **54.08** | +0.82 |
> > |           | 50%      | MoNE       | 48.91 | 24.74 | 44.96     | 23.10 | 45.81     | —     |
> > |           |          | MoNE+Step3 | 52.99 | 26.54 | 48.73     | 23.18 | **47.55** | +1.74 |
> > | Moonlight | 25%      | MoNE       | 80.89 | 50.94 | 58.88     | 49.25 | 64.33     | —     |
> > |           |          | MoNE+Step3 | 80.81 | 51.11 | 58.86     | 49.52 | **64.51** | +0.18 |
> > | Qwen      | 25%      | MoNE       | 80.35 | 54.27 | 59.52     | 72.83 | 68.83     | —     |
> > |           |          | MoNE+Step3 | 80.52 | 55.23 | 59.81     | 73.13 | **69.16** | +0.33 |
> >
> > Table 3 shows that step 3 provides modest but consistent gains (+0.18–1.74 points) to MoNE, validating that our bias updating strategy is generally beneficial. However, MoNE+Step3 still underperforms full STEP by 1.87 points on average (comparing with Table 2 in manuscript). This confirms that **STEP's gains arise from system-level synergy** combining token selection, dual-factor pruning, and bias updating, not solely from post-hoc tuning.
> >
> > ---

---

> > > ### Author Response · Authors · 2025-11-28
> > > **Response to Reviewer 6R1M (Part 3/3)**
> > >
> > > ## **W4: Multi-Token Generation and Long-Range Reasoning**
> > >
> > > We provide both theoretical and empirical justification:
> > >
> > > In causal Transformers, the final token's representation aggregates information from all preceding tokens via self-attention, **encoding the global context necessary for prediction**. Even for most multi-token generation methods[1–3], **generation proceeds autoregressively from the last token**. Each subsequent token conditions on previous ones, making the last token's attention pattern influential for the entire generation chain. We also evaluated STEP on HumanEval (code generation) and GSM8K (mathematical reasoning):
> > >
> > > **Table 4: Performance on generation tasks.**
> > >
> > > | Model               | Sparsity | Method    | HumanEval | GSM8K     | Avg       |
> > > | ------------------- | -------- | --------- | --------- | --------- | --------- |
> > > | **Qwen3-30A3B**     | Dense    | —         | 94.12     | 89.92     | 92.02     |
> > > |                     | 25%      | Frequency | 92.47     | 88.20     | 90.34     |
> > > |                     |          | HC-SMoE   | 92.48     | 87.49     | 89.99     |
> > > |                     |          | MoNE      | 93.09     | 88.17     | 90.63     |
> > > |                     |          | **STEP**  | **93.42** | **88.63** | **91.03** |
> > > |                     | 50%      | Frequency | 59.35     | 44.20     | 51.78     |
> > > |                     |          | HC-SMoE   | 34.15     | 54.97     | 44.56     |
> > > |                     |          | MoNE      | 63.78     | 41.09     | 52.44     |
> > > |                     |          | **STEP**  | **64.02** | **55.19** | **59.61** |
> > > | **Moonlight-16A3B** | Dense    | —         | 48.17     | 74.37     | 61.27     |
> > > |                     | 25%      | Frequency | 19.76     | 45.94     | 32.85     |
> > > |                     |          | HC-SMoE   | 23.17     | 49.40     | 36.29     |
> > > |                     |          | MoNE      | 27.07     | **53.98** | 40.53     |
> > > |                     |          | **STEP**  | **27.64** | 53.07     | **40.36** |
> > > |                     | 50%      | Frequency | 13.66     | 10.01     | 11.84     |
> > > |                     |          | HC-SMoE   | 9.24      | 4.32      | 6.78      |
> > > |                     |          | MoNE      | 17.93     | 27.22     | 22.58     |
> > > |                     |          | **STEP**  | **18.31** | **30.40** | **24.36** |
> > >
> > > At 25% sparsity, STEP maintains 98.9% of dense performance on Qwen (91.03 vs. 92.02), outperforming all baselines. At 50% sparsity, all methods degrade more severely on generation tasks than classification. This reflects error accumulation during autoregressive decoding that amplifies representation distortions from aggressive pruning. STEP achieves the best retention among methods, **gaining +7.83 points over Frequency and +7.17 over MoNE at 50% on Qwen**.
> > >
> > > These analyses and results demonstrate that our last-token attention approach generalizes effectively to long-range reasoning and multi-token generation scenarios.
> > >
> > > **References:**
> > >
> > > [1] Stern, M., Shazeer, N., & Uszkoreit, J. Blockwise parallel decoding for deep autoregressive models. NIPS 2018.
> > >
> > > [2] Gloeckle, F., Synnaeve, G., Cordts, B., Rother, C., Verbeek, J., & Larlus, D. Better & faster large language models via multi-token prediction. ICML 2024.
> > >
> > > [3] DeepSeek-AI. DeepSeek-V3 Technical Report. arXiv preprint arXiv:2412.19437.
> > >
> > > ---
> > >
> > > ## **Q1: Routing Changes After Pruning**
> > >
> > > We visualized routing distributions before/after pruning on C4, WikiText and ARC-Challenge datasets. We find that **routing patterns exhibit minimal divergence**, indicating that our bias updating step effectively compensates for expert removal **without destabilizing the routing mechanism**. These visualizations are included in the revised Appendix I on pages 20–21.
> > >
> > > ---
> > >
> > > ## **Q2: Performance Without Bias Updating**
> > >
> > > **Table 5: STEP performance without bias updating (Step 3). Average across 8 benchmarks.**
> > >
> > > | Model     | Sparsity | D2-MoE | HC-SMoE | MoNE  | STEP w/o Step 3 | STEP (Full) |
> > > | --------- | -------- | ------ | ------- | ----- | --------------- | ----------- |
> > > | OLMoE     | 25%      | 54.60  | 54.88   | 53.26 | 57.51           | **58.42**   |
> > > |           | 50%      | 46.88  | 43.69   | 45.81 | 48.31           | **50.05**   |
> > > | Moonlight | 25%      | 61.25  | 53.89   | 64.33 | 64.97           | **65.27**   |
> > > |           | 50%      | 50.88  | 42.99   | 54.47 | 54.82           | **56.20**   |
> > > | Qwen      | 25%      | 68.77  | 64.30   | 68.83 | 68.74           | **69.06**   |
> > > |           | 50%      | 56.30  | 55.54   | 64.15 | 65.05           | **65.78**   |
> > >
> > > Critically, **STEP without Step 3 already surpasses or matches all baseline methods** (including D2-MoE, HC-SMoE, and MoNE). This demonstrates that STEP's core contributions (token selection + dual-factor pruning) provide substantial gains even without bias updating, with Step 3 offering additional refinement (+0.91–1.74 points). This ablation confirms the robustness of our approach.
> > >
> > > ---

---

### Author Response · Authors · 2025-11-28
**Response to All Reviewers**

We thank all reviewers for their careful reading and constructive feedback. Your comments have helped us significantly strengthen the paper. Below we summarize the main concerns raised and our responses.

### **Major Concerns and Our Responses**

Several reviewers (R1-6R1M,R2-osTy,R4-sxUJ) questioned the **theoretical justification for using attention scores as token importance measures**. We have theoretically added a formal gradient-based derivation showing that attention weights naturally encode task-relevant importance through the optimization process. We also provide direct experimental validation through per-layer MSE analysis, demonstrating that our attention-guided token selection achieves 7–75% lower representation error compared to frequency-only approaches.

One reviewer (R3-Biuy) asked about the novelty of our **dual-factor pruning metric**. We want to clarify that **our core contribution is introducing token-aware importance analysis into expert pruning**, which prior MoE compression methods do not consider. The frequency × feature norm metric is intentionally simple and direct, **making STEP practical and broadly applicable**. We have added comparisons showing this simple metric outperforms more complex alternatives including Hessian-based approaches while requiring 130× less computation time.

Reviewers (R1-6R1M,R2-osTy) raised valid concerns about **calibration fairness and the role of bias updating**. We conducted extensive ablations showing that **STEP without any fine-tuning already outperforms all baseline methods**, confirming that our gains come primarily from token-aware expert selection rather than post-hoc calibration. We also applied our calibration procedure to competing method and found it provides modest improvements (+0.18–1.74 points), but **these enhanced baseline still underperform full STEP**, validating the synergy of our complete framework.

Several reviewers (R1-6R1M,R2-osTy,R4-sxUJ) requested evaluation on **generation tasks and larger models**. We have added comprehensive experiments on HumanEval (code generation) and GSM8K (mathematical reasoning), where **STEP maintains 98.9% of dense performance at 25% sparsity and achieves the best retention among all methods at 50% sparsity**. We also validated **scalability on GLM-Air (106B parameters)**, demonstrating consistent advantages on 100B+ scale models.

Reviewers (R3-Biuy,R4-sxUJ) questioned the theoretical basis of the bias compensation method. We respond that the bias vectors are derived from a **closed-form optimal solution minimizing squared error, ensuring layer-wise approximation**. This is followed by global fine-tuning using gradient descent to jointly optimize all biases across layers, achieving a global optimum on language modeling objectives. **Experimental results confirm they improve performance across methods without overfitting, validating their low-cost effectiveness**.

One reviewer (R3-Biuy) raised concerns about token distribution randomness in calibration. Our ablation studies across different sample sizes and sequence lengths show that **STEP achieves remarkably stable performance (variance <0.3 points) across configurations, whereas baseline methods show up to 3.1 point swings**. This validates that statistical aggregation over moderate sample sizes effectively captures stable importance signals despite sample-to-sample randomness.

### **Improvements to Paper Quality**

These revisions have substantially strengthened the paper. The theoretical grounding now provides rigorous justification for our design choices. The expanded experiments demonstrate broad applicability beyond classification tasks and scalability to 100B+ parameter models. The ablations clearly isolate the contributions of each component and show robustness to design choices. The fair comparisons and transparent reporting ensure reproducibility and build confidence in our results.

We believe our response specifically and satisfactorily addressed the weaknesses and questions you raised. We sincerely hope our revisions meet your approval and look forward to your positive response.

---

### Author Response · Authors · 2025-12-02
**Response to Area Chairs**

Thank you for overseeing the review of our submission in this challenging situation. We have carefully addressed all reviewer concerns during the rebuttal period and would like to briefly summarize our key responses to assist your evaluation.

### **Theoretical Justification and Experiment Validation for Attention-Based Token Selection** (R1-W1, R2-W3, R4-W2)

Attention-based token selection is widely used in prior work for identifying salient inputs. Our contribution is to show that pruning based on these important tokens more effectively reveals redundant experts. To provide direct evidence beyond this established practice, we added gradient analysis demonstrating how attention weights capture task relevance, and **per-layer MSE experimental validation confirming 7–75% lower representation error compared with frequency-only methods**.

### **Novelty (Metric Design, Similarity to Existing Frameworks)** (R3-W2, R4-W4)

Our frequency × norm pruning metric is intentionally lightweight for large-scale deployment, outperforming Hessian-based methods with **130× lower computational cost**. STEP's core difference is using token importance to determine which experts to prune, rather than optimizing reconstruction errors as explored in existing frameworks.

### **Calibration Fairness** (R1-W3, R2-W2)

We applied our bias-updating step (Step 3) to the competing baseline MoNE. While it brings +0.18–1.74 point improvements, MoNE + bias-updating still **lags STEP by 1.87 points on average**. Notably, **STEP without calibration already outperforms all baselines**, and calibration further boosts performance, confirming that each component contributes meaningful gains.

### **Generation Tasks & Scalability** (R1-W4, R2-W4, R4-W3)

We added HumanEval and GSM8K evaluations. **STEP retains 98.9% of dense performance at 25% sparsity** and achieves the best retention at 50% (+7.2 points over MoNE on Qwen). We also validated STEP on GLM-Air (106B), showing consistent advantages at the 100B+ scale.

### **Bias Compensation Theory and Robustness Analysis** (R3-W1, R3-W3)

We provided closed-form derivations for bias vectors that minimize layer-wise error with global gradient refinement. Cross-method and cross-dataset validation shows consistent improvements without overfitting. Sample-size ablations indicate that **STEP maintains stable performance (variance <0.3), while baselines fluctuate by up to 3.1 points**, confirming that 128 samples are sufficient for reliable importance estimation.

---

Additionally, all other issues have been addressed and clarified with experiments, as detailed in the rebuttal. The revision **strengthens theoretical foundations**, **extends evaluation to generation tasks and 100B+ models**, and **provides fair component-wise comparisons**.

---

### Note · Authors · 2026-01-29

I have read and agree with the venue's withdrawal policy on behalf of myself and my co-authors.

---

### Meta-Review · Area_Chair_HkJv · 2026-01-04

**Summary:**

The reviewers’ concerns that informed the decision can be summarized as follows:

1. Reviewers questioned whether the core ideas, attention-based token importance, frequency- and norm-based expert scoring, and lightweight calibration, constitute a fundamentally new insight or are incremental combinations of existing heuristics already explored in prior MoE pruning and compression work.

2. Several reviewers expressed concern that using last-token attention as a proxy for token importance is brittle, task-dependent, and not a universally reliable indicator of expert importance, particularly across diverse objectives or prompts.

3. Although presented as lightweight, the expert-to-bias conversion and calibration step raised questions about fairness and whether comparable post-processing could similarly improve baselines, potentially narrowing the reported advantage.

4. While improvements were observed, reviewers noted that gains over strong baselines (e.g., MoNE, HC-SMoE) are often modest, especially at lower sparsity, making it unclear whether the added methodological complexity is justified.

**Reviewer Concerns:**

Concerns addressed by the rebuttal:

1. The authors added gradient-based derivations and per-layer MSE analyses to support the claim that attention correlates with token importance. This addresses the request for stronger justification and improves the paper’s rigor.

2. Additional experiments applying bias updating to baselines and ablations removing Step 3 help clarify the contribution of each component and demonstrate that calibration alone does not fully explain the gains.

3. The inclusion of generation tasks (HumanEval, GSM8K) and larger-scale models (100B+) addresses reviewer requests for broader validation.

Concerns still outstanding:

1. Despite additional justification, the core methodology still appears as a careful engineering refinement of existing pruning heuristics rather than a conceptually new pruning principle. This concern was not fully alleviated by the rebuttal.

2. The rebuttal strengthens the argument but does not fully dispel concerns about brittleness or task-specificity of last-token attention as a universal pruning signal.

3. The empirical gains, while consistent, remain relatively small in many regimes, leaving unresolved whether the proposed framework meaningfully advances the state of the art compared to simpler or existing approaches.

**Reviewer Scores:**

Reviewer 6R1M (initial rating: 4  marginally below threshold):
Likely to still be below acceptance due to concerns about performance and robustness.

Reviewer osTy (initial rating: 6  marginally above threshold):
Likely remain around 6.  The probem seem to be addressed yet the performance boost is small.

Reviewer sxUJ (initial rating: 4  marginally below threshold):
Minor upward adjustment possible due to the thorough rebuttal, but unlikely to offset the fundamental concerns such as novelty.

Reviewer Biuy (initial rating 4 marginally below threshold):
Likely to still be below acceptance due to concerns about novelty.

---

### Decision · Program_Chairs · 2026-01-26

Reject